# Estimating the AMOC from Argo Profiles with Machine Learning Trained on Ocean Simulations

Yannick Wölker[1,2], Willi Rath[1], Matthias Renz[2], and Arne Biastoch[1,2]

[1]GEOMAR Helmholtz Centre for Ocean Research Kiel, Kiel, Germany
[2]Christian-Albrechts Universität zu Kiel, Kiel, Germany

**Correspondence:** Yannick Wölker (ywoelker@geomar.de)

**Abstract.** The Atlantic Meridional Overturning Circulation (AMOC) plays an important role in our climate system, continuous monitoring is important and could be enhanced by combining all available information. Moored measuring arrays like RAPID divide the AMOC in near-surface contributions, western-boundary currents, and the deep ocean in the interior of the basin. For the deep-ocean component, moorings measure density and focus on the calculation through geostrophy. These moored devices come with a high maintenance effort. Existing reconstruction studies show success with near-surface variables on monthly time scales, but do not focus on the interior transport. For interannual to decadal time scales, the geostrophic contribution becomes an important contribution.

Argo floats could provide required information about the geostrophic circulation as they continuously and cost-effective deliver hydrographic profiles. But they are spatially unstructured and only report instantaneous values. Here we show that the geostrophic part of the AMOC can be data-drivenly reconstructed by Argo profiles. To demonstrate this, we use a realistic and physically consistent high-resolution model VIKING20X. By simulating virtual Argo floats, we demonstrate that a learnable binning method to process the spatially variable Argo float distribution is able to reconstruct the geostrophic part of the VIKING20X AMOC by up to 80% explained variance and a mean error of less than one Sverdrup for the geostrophic transport. Using methods of explainable AI we investigate the importance of our input components showing an increasing importance of the Argo profiles on seasonal and interannual timescales, validating the usefulness of the Argo floats for the reconstruction. Our results demonstrate how an AMOC reconstruction from unstructured Argo profiles could replace estimates of the geostrophic deep-ocean component of the AMOC from the RAPID Array in the context of high-resolution ocean and climate models.

## 1 Introduction

The Atlantic Meridional Overturning Circulation (AMOC) is a key part of the climate system, and has specific socio-economic implications for Europe and North America (Zhang et al., 2019). It moves warm and salty water northward in the upper layers of the Atlantic Ocean, while colder and fresher waters flow southward at depth (Lozier, 2012). Owing to this vertical temperature contrast, the AMOC carries about 1.2 PW ($1PW = 10^{15}W$) in heat northward and regulates global and regional climate (Johns et al., 2023). The AMOC is usually described as the total overturning transport of water across a full section of the Atlantic. Because of its importance, its strength and variability are closely watched, and the question of how it might

change in the future is widely debated (Caesar et al., 2021; Latif et al., 2022; Terhaar et al., 2025; Baker et al., 2025). The AMOC has been studied using both observational data (McCarthy et al., 2012; Li et al., 2021) and numerical models (Biastoch et al., 2021; Petit et al., 2023). Both ocean and climate models often fail to accurately simulate the AMOC (McCarthy and Caesar, 2023), potential simulated AMOC trends are possibly due to artificial numerical choices such as those made to balance the freshwater budget (Behrens et al., 2013; Biastoch et al., 2021) or the inability of coupled climate models to simulate the correct decadal variability (Jackson et al., 2022).

As an integral quantity describing the combined effect of several current systems, the AMOC is constructed from several components representing different transport processes, including boundary currents, wind-driven Ekman transports at the surface, and slower-moving currents in the (deep) ocean interior (Moat et al., 2024; Frajka-Williams et al., 2019). Further complication comes from the fact that each of these processes is governed by different timescales. While the AMOC strongly varies on daily to seasonal timescales, with even short-term reversals, interannual changes mask the variability, the decadal variability and potential long-term trends of the AMOC are most important for climate (Latif and Keenlyside, 2011).

A major effort to monitor the Atlantic Meridional Overturning Circulation (AMOC) is undertaken at 26.5°N through the international RAPID project (Cunningham et al., 2007; Johns et al., 2023). At this latitude, RAPID breaks the total overturning down into three main parts: First, the wind-driven Ekman transport measured by satellites (McCarthy et al., 2015). Second, the strong Florida Current at the western boundary monitored by submarine telephone cables between Florida and the Bahamas. Measurements of changes in electrical resistance provide a continuous time series at a daily resolution (Larsen, 2001). These two components are routinely available and can be directly included to construct the AMOC. The third part, the upper mid-ocean transport, is more difficult to obtain. The mid-ocean transport is separated further into the western boundary wedge (Johns et al., 2008), and the interior geostrophic transport, estimated from moorings at the eastern and western boundaries (McCarthy et al., 2015). In addition, a barotropic transport term is included as a closure to ensure net-zero mass transport across the section. Establishing and maintaining such a mooring system is costly and depends on major international collaboration. Since measurements of the ocean interior are limited in time and location, the question arises of whether the geostrophic transport in the ocean interior can be estimated with less effort and data that is more widely available, and also for other latitudes.

This study explores the concept of using Argo floats, as an alternative for measuring the water mass structure and the derivation of geostrophic gradients in the upper 2000 meters (Wong et al., 2020; Roemmich et al., 2009; Argo, 2025). Argo floats offer a cost-effective way to observe the ocean interior. They have the potential to replace the upper part of the RAPID moorings. The Argo float cycle involves drifting at a depth of around 1,000 meters and a 2000 meter dive before returning to the surface. Argo floats have strong potential for estimating basin-wide transport because they provide continuous hydrographic data across the upper ocean. However, their irregular timing and spatial coverage make it challenging to apply standard reconstruction methods without information loss.

Earlier attempts to estimate the AMOC often relied on satellite-based measurements of surface quantities such as sea surface height (SSH), wind stress, or on gravimetric estimates of the Ocean bottom pressure (OBP). How well these methods perform generally depends on the timescale considered. For example, Frajka-Williams (2015) estimated the AMOC at 26°N using SSH together with estimates of the Florida Current and Ekman transport, capturing up to 90% of the interannual variability. However,

unlike the RAPID approach, Frajka-Williams and others like Sanchez-Franks et al. (2021) do not use direct measurements but rather quantities which are observable at the surface to approximate the ocean interior. Other studies have included subsurface data to improve the reconstruction of geostrophic transport. For instance, Solodoch et al. (2023) showed that bottom pressure and zonal wind stress in the ocean reanalysis ECCOv4 (Forget et al., 2015) are suitable to reconstruct the AMOC at this latitude using a simple neural network. However, since bottom pressure measurements are not widely available, Argo floats appear to be an attractive alternative, even though they do not deliver direct transport measurements. Multiple studies were conducted to average Argo profiles in regular cells (Zilberman et al., 2020; Willis, 2010), from which dynamic height fields can be used to estimate interior geostrophic transports (McMonigal et al., 2018; Holte and Straneo, 2017). A comparison between the estimated Argo transports and the OBP gradients at the western continental shelf was performed by Elipot et al. (2014). So far, these approaches rely on spatial and temporal binning to organize the Argo data, and average over multiple profiles to produce grid-based fields for further computation. Typically, three-month windows and spatial binning (e.g., quarter-degree) is used, which can smooth out important details and reduce the value of individual profiles. None of the mentioned approaches fully preserves the detail in individual profiles, especially on short timescales where data sparsity becomes a major challenge. There remains a gap in combining modern machine-learning approaches (Solodoch et al., 2023) with Argo float data.

Machine learning methods for reconstructing the AMOC rely on supervised learning, requiring paired input-output data for training. Even though the RAPID array has been collecting data for almost 20 years (Johns et al., 2023), these data are not enough to train a reliable reconstruction, especially when targeting timescales longer than a month. For example, splitting the 20 years of RAPID data into 90-day averages would only yield around 80 independent samples. Hence, observational data alone cannot be used. Since observational datasets are limited in their coverage in time, we can not rely on them to assess the generalizability to periods beyond the training data. To work around the issue of data scarcity, earlier studies (Solodoch et al., 2023; Sonnewald and Lguensat, 2021) used ocean reanalysis products like ECCOv4. Ocean reanalyses and ocean general circulation models provide longer and fully spatial and temporal covered datasets for training. While these ocean models do have their own biases and uncertainties, they are generally consistent in terms of the underlying physical principles formulated in the hydrodynamic equations and how different parts of the AMOC respond to surface forcing (Biastoch et al., 2021; Böning et al., 2023). In particular, high-resolution ocean models have been proven to simulate the important aspects of the wind-driven and thermohaline circulation, including western boundary current and mesoscale dynamics, deep formation and spreading of water masses (Hirschi et al., 2020) This makes them a solid basis for learning heterogeneous ocean states with inputs like wind stress, Florida Current, Antilles Current, and hydrographic conditions (see Figure 1). Despite remaining errors due to numerical choices and parameterizations, models can be used as a well-defined testbed to develop and evaluate machine learning approaches for AMOC reconstruction. They offer a controlled but realistic setting in which the data quality can be similar to the real world observations but with a increased temporal extent.

In this work, we demonstrate how and to what extent the AMOC can be reconstructed in an ocean model setting from simulated measurements that mimic widely available observational products. Reconstructing the AMOC on different time scales from 10 days up to five years, we find that the importance of geostrophic transport in the ocean interior becomes more pronounced, with longer time scales. To capture this, we use virtual Argo profiles as our observational input, leveraging their

insight into the ocean interior while addressing their sparse and irregular sampling with a graph-based neural network approach. To overcome the irregular spacing, we adapt SUSTeR (Wölker et al., 2023), an approach for reconstructing spatio-temporal events from unstructured sparse observations spread over a wide area using graph neural networks. We extend this approach, originally developed for traffic reconstruction, to handle drifting sensors like Argo floats. We train and evaluate our method with virtual Argo floats and virtual deep RAPID moorings sampled from the high-resolution ocean simulation VIKING20X (Biastoch et al., 2021).

While our main goal is to train a reconstruction for the AMOC at 26.5°N based on direct measurements of Florida Current transport, Antilles Current transport, wind stress estimates, and on Argo float and mooring data, we also investigate other experiments validating our framework by highlighting different components of the AMOC. To this end we not only take the AMOC from VIKING20X as training target, but additionally generate a RAPID-like interior geostrophic transport time series from virtual moorings. Our method successfully reconstructs the AMOC at 26.5°N on seasonal and longer timescales, largely due to its ability to recover interior geostrophic transport from Argo float profiles.

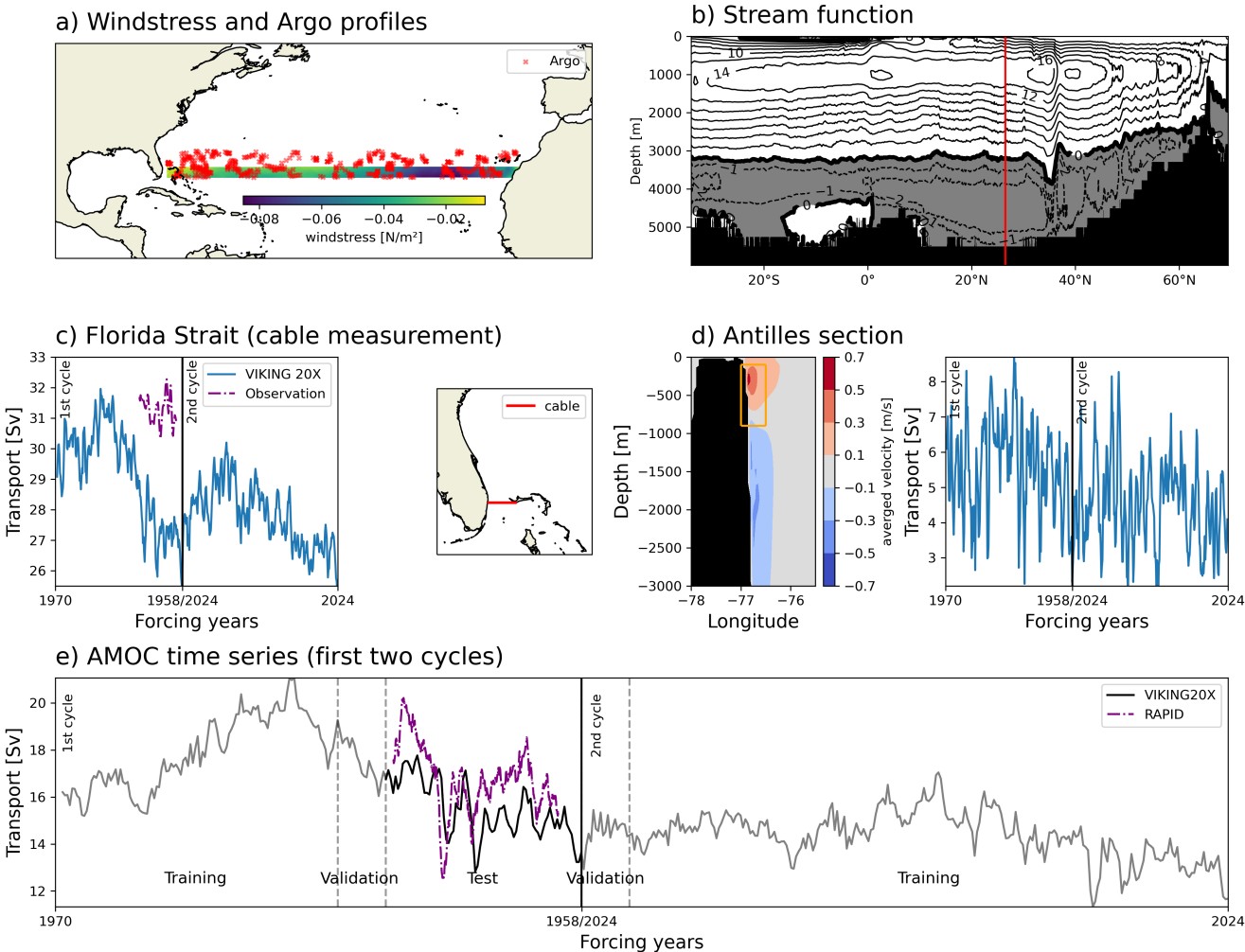

**Figure 1.** Sketch of all modalities that are included in the reconstruction of the AMOC in the context of the VIKING20X simulation. a) shows where wind stress around the target latitude was used for a sample season in 2015 together with all Argo position that were available for this 90 days period. The overturning stream function in the Atlantic basin b) in the reconstruction period from 2005 to 2022 shows the target latitude with a red line and the average structure of the overturning. The panel c) and d) show the Florida Current and the Antilles Current respectively. The left sub-panel shows the location of the measurement and the right sub-panel shows the resulting seasonal-mean transport time series used as input for the reconstruction. e) shows the AMOC time series along the 26.5°N latitude from the first two cycles in the VIKING20X simulation which we use as training target for the reconstruction.

## 2  Data

### 2.1  Simulation Cycles of the eddy-rich ocean model VIKING20X

VIKING20X is an ocean general circulation model configuration with a high-resolution nest in the Atlantic basin. The nest provides eddy-rich resolution not only for the region of interest around the latitude 26.5°N but also at all relevant key regions for AMOC change. The model configuration is based on the NEMO framework and has a global OCRA025 configuration (1/4° resolution) with a higher resolution nest (1/20° resolution) covering the Atlantic from 33.5°S to about 65°N using Adaptive Grid Refinement (AGRIF) (Debreu et al., 2008). In both, the base and the refined nest component, the configuration has 46 vertical z-levels with a vertical resolution decreasing from 6m close to the surface to 250m close to the ocean floor. The model is subject to the JRA55-do atmospheric forcing between 1958 and 2023 (Tsujino et al., 2018). In alignment with the OMIP-II protocol (Tsujino et al., 2020), six cycles of the model simulation each covering the period of 1958 to 2023 were sequentially executed with the same forcing for each cycle. The first cycle was initialized from an ocean at rest and from hydrographic conditions reflecting the World Ocean Atlas 2013 (WOA13, Locarnini et al. (2013), Zweng et al. (2013)). Subsequent cycles were started from the last time step of the previous cycle with a restarted forcing period. To account for a spinup time of the model, we only used the first cycle from 1970 onward. The high-resolution simulation VIKING20X is known to resolve mesoscale processes in the entire North Atlantic which includes key processes for the variability of the AMOC such as deep water formation and the process of deep convection observable in the subpolar North Atlantic (Rühs et al., 2021), the distribution of fresh water in the subpolar gyre (Fox et al., 2022), and the overflow pattern (Böning et al., 2023), as well as aspects of the wind-driven circulation such as western boundary currents and mesoscale eddies (Biastoch et al., 2021).

The AMOC strength and variability in the first cycle is described in detail in (Biastoch et al., 2021) and is used in this study as the reconstruction target. To increase the amount of data that can be utilized for the training process, we also use virtual measurements from the second cycle. The AMOC strength for the first two cycles is shown in Figure 1 e) on a seasonal time scale with 15.5Sv mean transport and 2.6Sv standard deviation. While the first of the VIKING20X cycles has a stronger decadal AMOC variability associated with the deepwater formation in the 1990s in the subpolar North Atlantic (Böning et al., 2023), the second cycle simulates a reduced decadal AMOC variability (Schiller-Weiss et al., 2025). We expect that the generalizability of our trained reconstruction would profit from an increased heterogeneity of the training data as long as the training data and training targets represent consistent physical relations. Hence, even though the weakening decadal variability of the AMOC over the course of the six OMIP cycles may be due to spurious model drift, it may still be an advantage to include data from all six OMIP cycles into our training.

### 2.2  RAPID array

The RAPID Array is a multi-national effort to monitor the overturning at 26.5°N and is deployed since April 2004. (Kanzow et al., 2008; McCarthy et al., 2015; Frajka-Williams et al., 2019) RAPID estimates the AMOC in three different contributions, the Ekman transport with remote sensing product (McCarthy et al., 2015), the Florida Current transport via an underwater cable in the Florida Strait (Larsen, 2001), and the upper mid-ocean component. This upper mid-ocean can be divided into

the Antilles Current measured by moored current meters (Johns et al., 2008) and the interior geostrophic transport which is measured by moorings over large distances between the continental margins (Kanzow et al., 2008; Sinha et al., 2018). We will use the interior geostrophic transport calculation from RAPID in the context of the ocean simulation to investigate how much geostrophic variability can be reconstructed on different time scales with Argo floats.

### 2.3 Argo floats

Argo is an international network of Lagrangian floats which reside and drift with the ocean currents at 1000m depth and profile the upper 2000m of the water column in a fixed interval of 10 days (Riser et al., 2016). Argo floats provide the methodology to measure the interior of the ocean without any fixed mooring or recurrent ship survey. However, coming with the cost of a sparse and unstructured spatial distribution. Argo floats were first deployed in 1999 and their coverage has continuously grown until 2012 with approximately 430 Argo profiles per season in our study area.

Previous works averaged Argo measurements along larger time frames, Willis (2010); Hernández-Guerra et al. (2010) creating zonal-depth section of temperature and salinity profiles from which the geostrophic transport is inferred. Also the velocity information of the Argo floats at their parking depth, estimated by the offset of the surface position, was used to estimate transport (Reeve et al., 2019; Zilberman et al., 2020, 2023). Those methods neglect any individual float correlations due to the limitation of unstructured profiles. Solodoch et al. (2023), who reconstructed the AMOC with bottom pressure used for information about the ocean interior, highlighted the need to investigate how Argo floats can be adopted for this task. To handle the spatial distribution of Argo profiles, we here use a graph data structure (Dale and Fortin, 2010), which allows for a flexible topology.

### 2.4 Input & Output data for AMOC Reconstruction

While this study investigates to which extent observation strategies from the RAPID mooring array can be explained by Argo float data, it cannot directly use the observations from the real ocean, because the RAPID array only covers a period of 20 years. The resulting amount of independent samples from 20 years of observational data is not sufficient for training our machine learning framework, on longer than monthly or seasonal timescales. A supervised machine-learning approach is a general function approximation which requires data samples containing an input, the widely available measurements, and a target variable, the interior geostrophic or AMOC transport, to learn the reconstruction. In the following sections and summarized in Fig. 2 we describe in detail how the input and output is extracted from the ocean simulation.

**Observation-like input products**

The AMOC reconstruction is based on zonal wind stress, Florida Current transport, Antilles Current transport, Argo profiles for the upper 2000 meters, and the RAPID mooring information below (all components shown in Figure 1). To estimate the wind stress which in the RAPID approach is based on a remote sensing product, we take the zonal wind stress field from the forcing (calculated through Bulk formulae from the velocity difference between wind and surface ocean velocity) between 25°N and

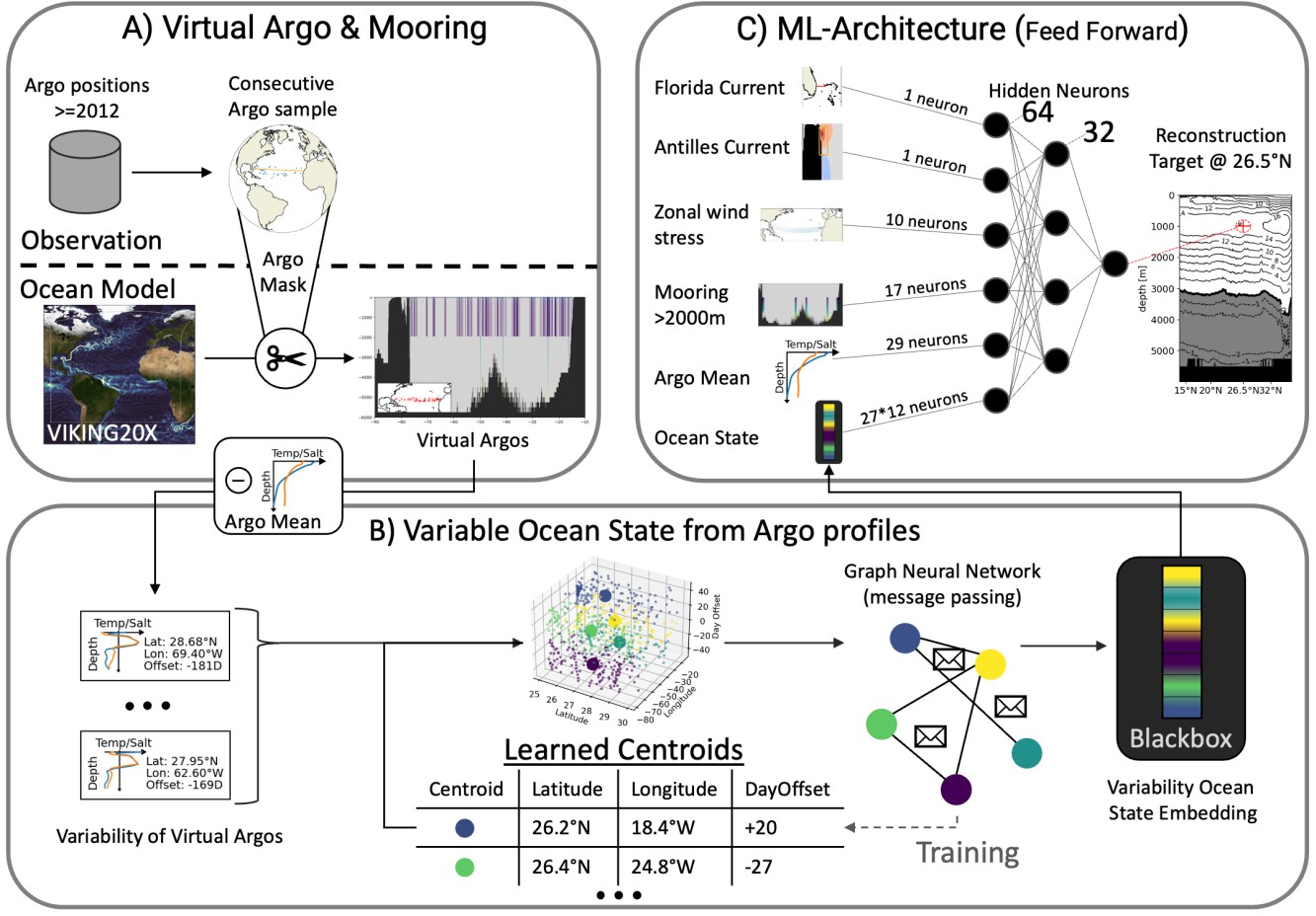

**Figure 2.** The workflow of the AMOC reconstruction with Argo profiles. A) shows the process of creating virtual Argo profiles from observational positions and relative times but with data extracted from an ocean simulation. The Argo profiles are divided for each time window in the mean profile and the residuals which are both used separately from each other. The variability is used in part B) of the figure to estimate the variable ocean state which is an embedding vector, also interpreted as black box, due to the end-to-end training. The reconstruction utilizes the strong power of GNNs to exchange information between learned cluster centers that aggregate single Argo profiles. The Argo mean, the estimated ocean states, and the inputs from Figure 1 are used in a feed-forward neural network to reconstruct the strength of the AMOC at 26.5°N and 1000 meter depth.

27°N (Figure 1 a)) and create 10 zonal bins. Although the Ekman transport could be estimated based on zonally averaged zonal wind stress alone, we provide 10 bins of averaged zonal wind stress to retain some information about the spatial structure. The transport through the Florida Strait was calculated as the integrated volume transport that is discussed in further detail in Biastoch et al. (2021) for the first OMIP cycle of the ocean model VIKING20X. It is consistent with the observations (about 2Sv or 7% weaker) and interannually correlated ($\sim 0.75$) with the AMOC. Inputs for the upper mid-ocean RAPID component are split into the Antilles Current and the interior of the ocean. For the Antilles Current we utilize a small box based on the mean transport pattern in VIKING20X in figure 1 d) from which we extract the Antilles Current transport. For the first two cycles in the Figure 1 d) the Antilles Current has an average transport of 5.5 Sv and 2.5 Sv standard deviation for seasonal time scales. This is meant to summarize the dynamics in the ocean state for everything that is west to the WB3 mooring in the RAPID array marking the boundary between the interior transport and the western boundary current. (Johns et al., 2011) For the region east of mooring WB3, we investigate the suitability of the Argo profiles as an alternative for the RAPID moorings in the upper 2000 meters. The density of the virtual Argo profiles are used in the Argo processing module of the reconstruction with the goal to identify the observed ocean state. Additionally, we provide the stream function below 2000 meters calculated from virtual RAPID moorings as an input to the reconstruction. For the calculation of the stream function we used a level of no motion at 4820 meters, similar to the RAPID calculation, and integrated the transport from this bottom level up to 2000 meters depth. Please note that the inputs to our neural network do not require to have the same physical units, since the learning process of the neural networks is based on the statistics of the input variables and not on their semantics. In section 4.4, we investigate the effectiveness of deep information by first testing the removal of the additional mooring data. Next, we assume that all Argo floats would reach up to 6000 meters.

We call the Argo floats and moorings, that we extract from VIKING20X, *virtual* to make clear that we mimic the spatial and temporal distribution but take the values from the simulation output (Figure 2 A). Therefore, we extract the horizontal grid box of the VIKING20X simulation nearest to the real Argo location from the daily output. To ensure the virtual profiles have the same spatio-temporal characteristics (and deficits) as the real Argo profiles, their position and time is sampled from the real-world Argo distribution. One example of such deficits is that consecutive real-world profiles from a single float are not independent from each other and that hence a uniform sampling across the Atlantic would grant too many degrees of freedom. To mitigate this we take random but consecutive time windows from the real world Argo distribution. The corresponding real-world time window of Argo profiles is sampled with uniform probability from all time windows of the respective length in the Atlantic between 25°N and 30°N, from 2012 (when the Argo array was full established) until the end of our simulation period 2023.

The processed virtual Argo product then provides these variables at vertical intervals of 20 dbar, as it is the standard for the real-world Argo floats. The ocean model from which we sample the virtual profiles has a higher resolution at the surface (starting with 6m-thick grid cells) and coarser resolution at depth (208m at 2000m depth). We linearly interpolate the 46 z-levels of VIKING20X onto the depth resolution of the real-world Argo profiles between 20dbar to 2000dbar. Since Deep Argo (Roemmich et al., 2019) increases the number of floats ranging to depths close to the ocean floor, we also include a section 4.4 where we sample virtual Deep Argo profiles.

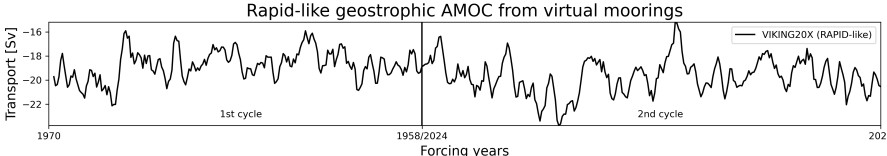

**Figure 3.** Showing the complete time series of the RAPID-like interior geostrophic transport from virtual moorings extracted from the ocean model VIKING20X. This data is the target variable for the *geostrophic* experiments and shown here with a season low pass filter(90 days).

**AMOC strength target time series (Model & RAPID-like)**

In this study, we investigate the reconstruction performance with two different ground-truth AMOC time series which are both calculated as the strength of the stream function at the grid box closest to 1000m in the ocean model. First, we use the zonally integrated AMOC transport, calculated from a basin-wide cross section of the meridional velocities. This time series will cover all three main RAPID components (Ekman, upper mid-ocean, and Florida Strait) (Frajka-Williams et al., 2019). Second, we put a strong focus on the predictive skill of Argo profiles for the interior geostrophic transport (Figure 3) and their potential to be an alternative strategy for moored instruments. For the interior geostrophic transport, we calculate the RAPID upper mid-ocean transport and exclude the western boundary wedge, therefore this is similar to the gyre recirculation in Smeed et al. (2018). To be as similar as possible to the RAPID interior transport, we took the RAPID calculation for the RAPID moorings (McCarthy et al., 2015) and mapped this to the VIKING20X output to share as many assumptions as possible. For a detailed description we refer the reader to appendix A.

**Time scales**

The AMOC is an integrated variable that compiles a number of physical processes on different time scales. For further insights, we perform the AMOC reconstruction on different time scales from very short time scales of 10-days, with expected dominance of the geostrophic turbulence, through yearly timescales dominated by the wind-driven circulation, and beyond to observe the shift from the wind-driven variability to the buoyancy-driven ocean transports (Kanzow et al., 2010). The used range of timescales for each of the experiments, as well as the input parameters and the target variable for the training and testing period is documented for overview in Table 1. For a given time scale, the AMOC strength, the Florida Current, Antilles Current, and the binned zonal wind stress, are averaged to the target time interval based on daily ocean model output (e.g. the monthly averaged time series in Figure 1). The virtual Argo profiles representing instantaneous measurements and can not be temporally averaged. Hence, we concatenate them into larger sets when the time scale is increasing.

## 3 Method

The machine-learning based reconstruction of the AMOC averaged to the respective target time scale, can be summarized as learning a set of parameters, represented by a trained reconstruction model (in the following referred to as trained recon-

| Experiment | Input | Output | Scales | Research question |
|---|---|---|---|---|
| Reconstruction (Exp.1) | FC, AC, ZW, AR + MO | AMOC | 10d-1y | To which extent can the ground truth AMOC of VIKING20X be reconstructed? |
| Extended reconstruction (Exp.2) | FC, AC, ZW, AR + MO | AMOC | 90d-5y | Is the method able to generalize to an unseen cycle? |
| Component importance (Exp.3) | All but one: FC, AC, ZW, AR + MO | AMOC | 10d-1y | How does the impact of each input changes with the time scales |
| Geostrophy Isolation (Exp.4) | FC, AC, ZW, AR + MO | RAPID-like (geos) | 10d-1y | Is the method able to reconstruct the geostrophic signal? |
| Deep Argo (Exp.5) | FC, AC, ZW, and one of: AR, DAR, or AR+MO | AMOC | 10d to 1y | Which kind of information is needed below 2,000 m? |

**Table 1.** For each experiment this table lists the used input features (FC - Florida Current, AC - Antilles Current, ZW - Zonal wind stress, AR+MO virtual Argo profiles <2,000m depth and virtual moorings stream function >2,000m depth, AR - only virtual Argo profiles, DAR - virtual Deep Argo profiles ) and which transport signal was set as the target either the AMOC from Figure 1 e) or our self-computed RAPID-like interior geostrophic transport in panel f)). The time scales on which the experiment was evaluated. Additional the short research questions summarize the line of arguments in this work.

struction), which uses a time-averaged zonal wind stress, a time-averaged Florida Current, a time-averaged Antilles Current, and a set of Argo profiles for the same averaging period, to reconstruct the strength of the time-averaged AMOC or interior geostrophic transport at the depth of 1000 meters. The overall workflow of the method is shown in Figure 2 with A) depicting the creation of virtual Argo profiles as described in section 2.4, B) showing the processing of this variable set of Argo profiles into a fixed-sized vector representation, and C) utilizing the fixed-sized vector representation from step B) together with directly used estimates of the other contributions to estimate the AMOC strength. The architecture of the neural network has a dedicated Argo processing module (Figure 2 B), because the mapping from a variable amount of spatially unstructured Argo profiles into a fixed-sized vector (black box in Figure 2 B) requires a more advanced method thatn for the other already structured contributions. We utilize artificial neural networks as our machine-learning framework to learn the reconstruction task due to their ability to approximate non-linear correlation in the unstructured spatio-temporal Argo profile distribution.

## 3.1 Processing of Argo profiles

Argo profiles pose a demanding challenge for the design of neural networks, as neural networks are normally designed to handle structured inputs. For spatio-temporal data points, structured input requires a constant number of measurements from constant locations with a static topology (e.g. a grid structure, or a graph), which is both not true for moving Argo floats. The virtual Argo profiles provide spatial information in form of longitude and latitude, and temporal information in form of a single point in time at which they reach the surface. A static clustering on all data points, could create such a structured input. Around our

targeted latitude this would amount to a zonal binning (Willis, 2010; Hernández-Guerra et al., 2010) but the set of Argo floats, especially for the shorter target time scales, show a heterogeneous distribution that would require carefully hand-crafted cluster boundaries. A classical binning comes with even more assumptions about shape, distance, and connectivity of the bins, which would require a thorough testing of hyper parameters. However, we aim in this work for a data-driven mapping function which identifies a structured numerical representation (n-dimensional vector) of the underlying ocean state which is independent from specific spatial distribution of the Argo profiles. Often, the term embedding is used for such a n-dimensional vector, but also for the vector space, and for the mapping (Meilă and Zhang, 2024). Constructing a suitable embedding (see black box in Figure 2 B) has two main aspects. First, the same ocean state observed by different spatial and temporal distributions of Argo profiles should result in the same (i.e. negligible difference) embeddings. Second, two embeddings should be similar (i.e. small difference) if the corresponding ocean states are similar, too. A successful embedding creates a vector representation of the manifold of ocean states that can be used to generalize for unseen but similar ocean states in the evaluation.

Therefore, we utilize a framework that learns a dynamic spatio-temporal clustering to create structured inputs. The framework Sparse Unstructured Spatio Temporal Reconstruction (SUSTeR, Wölker et al. (2023)) was originally developed for road traffic reconstruction from sparse observations in varying locations. SUSTeR set out with the goal to handle unstructured traffic observations and find a general representation of city traffic, much like the unstructured virtual Argo profiles with the goal to find a general representations of the ocean state. For technical details we refer the reader to the original paper, and provide in the following a brief intuition for the oceanographic audience. This framework acts as a learnable clustering with a fixed set of predefined clusters. The assignment decision is non-linear and adds each virtual Argo profile to one cluster based on its spatial and temporal component. A learnable clustering in contrast to the often used static clustering has dynamic boundaries between clusters which can represent different spatial and temporal resolutions, can have non-linear boundaries, or can change with global context (e.g. seasons). After each virtual Argo profile is assigned to one of the Learned Centroids (see Figure 2 B) the measurements of the virtual Argo profiles are aggregated within each cluster. These clusters represent the structured input, because their boundary will be fixed after training and the amount of clusters is constant. To learn the spatial and temporal correlation between these dynamic clusters the framework SUSTeR uses graph neural networks (Jiang and Luo, 2022), which can be thought of as message passing between the clusters, to learn the spatio-temporal correlations. After the clusters exchanged information, all features from the graph nodes (learned clusters) are concatenated and reduced in their dimensionality by a shallow feedforward neural network. The result is not human interpretable but serves as an embedding representing the ocean state.

## 3.2 AMOC strength reconstructed with Neural Network

The output of the Argo processing module is an embedding which is one input of the reconstruction module. We highlight the fact that the mean of all virtual Argo profiles in a single sample is subtracted from the profiles. The mean is separately used in the reconstruction module (see Figure 2 A), as the interior transport is characterized by a zonal pressure gradient, independent from the mean profile.

Together with the ten zonal wind stress bins, the Florida Current transport, the Antilles Current transport, and the mooring information below the 2000m Argo depth, the data are concatenated into an input vector for a feed-forward neural network. A feed-forward neural network is a sequence of layers containing a linear mapping and a non-linear activation function. Such an activation layer is the Rectified Linear Unit (ReLU) which is a composite function of the two linear function (constant zero for any negative values and a identity function otherwise). Feed-forward neural networks represent the basic form of neural networks to find correlations in data that do not use information about the hidden neighboring structure of the data, i.e. if the data are to be understood as sequential ordering, as a regular structured image, or as a more flexible topology. Such networks were already used to reconstruct the AMOC in the work of Solodoch et al. (2023). We found that our architecture requires more neurons compared to that of Solodoch et al. (2023) to handle the complex structure of the Argo processing vector representing the interior ocean state. The network is trained with backpropagation minimizing the mean squared error between the reconstruction and the ground truth interior geostrophic or AMOC transport, respectively. All input and output values are normalized with their respective mean and standard deviation for a faster training (Ioffe and Szegedy, 2015).

Further, we discuss the choice for the test, validation, and training periods of our data as it can be seen in Figure 1 e). While the data in the training period is used to iteratively train the reconstruction, the data in the validation period is used to determine when to stop the training. All evaluations are only based on the data in test period. With the data source VIKING20X we decided to use 2004 - 2023 from the first OMIP cycle as the test period for the reconstruction because those results compare well with the RAPID time series and other key components of the AMOC in respect to the statistics and temporal evolution (Biastoch et al., 2021). As the task of the validation period is to hinder the leakage of training data into the test period, it should be adequately sized depending on the underlying temporal autocorrelations. We will investigate up to interannual signals, therefore we set aside five-year windows before and after the test period as validation data. While the earlier validation period 1999-2004 is trivial, the second validation period of five years starts with the second forcing cycle. But since the ocean state is continuing into the next cycle, we expect the later validation period to break the auto correlation of the interior ocean state in the same way the earlier period does.

## 3.3 Evaluation metrics

To evaluate the skill of the AMOC reconstruction in comparison to the VIKING20X AMOC, we use four different metrics, the mean absolute error (MAE), the root mean squared error (RMSE), the mean absolute percentage error (MAPE), and the coefficient of determination (R2). For a comprehensive overview we define those skill metrics in the following for a test period with $N$ samples, $y_i$ being the targeted AMOC, and $\hat{y}_i$ being the reconstruction.

$$\text{MAE} = \frac{1}{N} \sum_{i=1}^{N} |y_i - \hat{y}_i| \tag{1}$$

$$\text{RMSE} = \sqrt{\frac{1}{N} \sum_{i=1}^{N} (y_i - \hat{y}_i)^2} \tag{2}$$

$$\text{MAPE} = \frac{1}{N} \sum_{i=1}^{N} \frac{|y_i - \hat{y_i}|}{y_i} \tag{3}$$

$$\text{R}^2 = 1 - \frac{\sum_{i=1}^{N}(y_i - \hat{y_i})^2}{\sum_{i=1}^{N}(y_i - \overline{y})^2} \tag{4}$$

Large outliers are less visible in the MAE compared to the RMSE where errors are squared. The difference between MAE and RMSE gives an insight whether the reconstruction is able to keep a certain reconstruction quality throughout the test period. Both values are reported in the same unit as the original time series (Sverdrup ($1Sv = 10^6 \frac{m^3}{s}$)). Because the interior geostrophic and AMOC transport have different amplitudes the relative MAPE allows for an comparison their reconstruction skills. The R2 score reports the amount of variability reconstructed from the target time series with 0% being a constant mean predictor and 100% a perfect reconstruction. As we work with neural networks the skill is dependent on the initial random initialization of the weights. In order to produce reliable results we execute each experiment eleven times and report the mean execution skills.

## 4 Results

### 4.1 Reconstruction of the AMOC at 26.5°N

To assess the reconstruction performance across a range of dynamical regimes, we evaluate AMOC reconstructions on timescales from 10 days to one year (Exp.1 in Table 1). Different drivers of the variability play a role on different time scales, which must be investigated to determine on which of the components, the reconstruction performs well and where it shows deficits. Short time scales are known to be driven by the atmospheric forcing in the study area, as shown by Kanzow et al. (2010), which enabled the AMOC reconstruction from Solodoch et al. (2023) to perform well on monthly data. They found a strong focus of the reconstruction at the wind stress, but they kept the monthly reconstruction and low-pass filtered it to create longer time scales. Here, we show that an already low-pass filtered input will shift the focus onto the density structure, hence increasing importance of the Argo component.

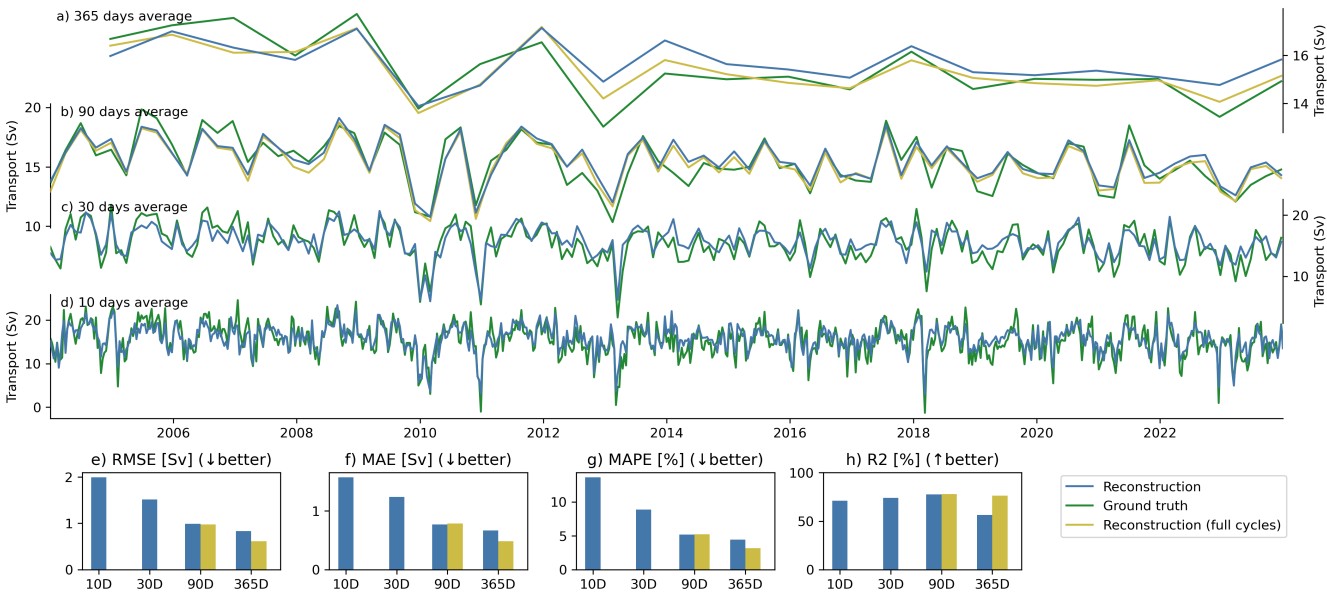

**Figure 4.** AMOC Reconstruction of the VINING20X AMOC, trained on the first two cycles except for the test period from 2004 to 2023 of the first cycle and the validation of five years on each side of the test data for yearly a), seasonal b), monthly c), and 10-day timescales. The VIKING20X AMOC is depicted in green, the reconstruction in blue, and the yellow reconstruction is trained with the additional cycles three to six. The skill of the reconstruction is measured in RMSE e), MAE f), coefficient of determination ($R^2$) g), and MAPE h).

The reconstruction generally captures AMOC mean and variability across all timescales (Figure 4), with decreasing accuracy on shorter time scales due to larger amplitudes. However, the short timescales not only show the largest mean absolute error (Figure 4 f)), also the largest relative error h) with over 10% mean error on average. We argue that those very short time scales

exhibit strong noise, which here we define as variability that is not represented in our input data. Examples for unexplained and therefore not reconstructed fluctuations on the 10-day time scales are changes in the spatial extent of the Antilles Current outside of our bounding box, a sparse amount of Argo profiles that not cover the boundaries or other fine scale processes. For the latter, mesoscale eddies and Rossby waves are potential signals that change the AMOC on such short timescales but are not resolved sufficiently in the input. This shows a limitation of the reconstruction approach for time scales of 10 days

or shorter. Despite the limits of the reconstruction on short timescales, we highlight that major features like the weak-AMOC events in end of 2010 or in the beginning of 2013 are well reconstructed (Biastoch et al., 2021). Short-term fluctuations are well captured, although the shortest 10-day time scales demonstrate a slight underestimation at the beginning of the test period (2005-2007) and show reduced variability at a smaller scale in the subsequent years (2019-2021). Most of the skill is lost in the small variations which is the reason for a worse MAE and RMSE compared to a moderate skill decrease in the R2 compared

to longer time scales.

Monthly reconstruction (Figure 4 c)) benefits from reduced high-frequency noise and shows a stronger signal-to-noise ratio. Compared to the 10-day timescale, all skill metrics improve for the monthly reconstruction. The signal-to-noise ratio improves

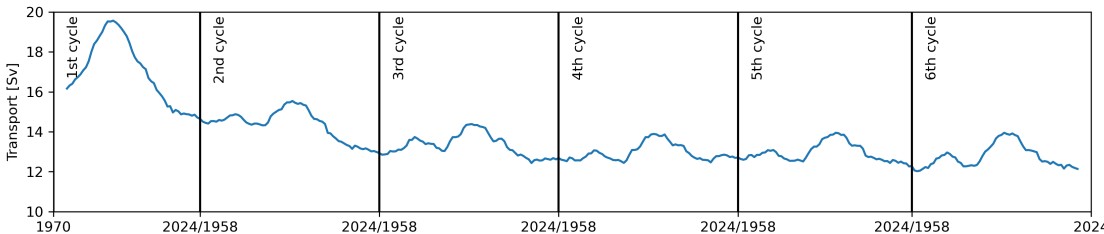

**Figure 5.** The AMOC strength for all consecutive six OMIP cycles of the ocean simulation VIKING20X. A 10-year low pass filter was applied to the timeseries. The vertical bars show the boundaries of the OMIP cycles and the x axis indicates the reoccurring forcing years.

for the selected input variables used in the reconstruction, while we achieve an explanation of 74% of the overall variance (R2 score). It also shows robust performance, as is apparent from the standard deviation of MAE and RMSE across eleven runs
with random seeds, which remains below 0.04Sv.

### 4.1.1   Extension to seasonal and yearly time scales

While the skill continues to improve on seasonal scales (Figure 4 b)), yearly reconstructions a) are constrained by limited training data. It is known that neural networks need vast amounts of training data to approximate the correlations in the data (Jiang and Luo, 2022; Ioffe and Szegedy, 2015) and exploit them for well performing reconstructions. At the yearly
scale, each training sample corresponds to one year of data. Originally, we use years 1970–2024 from the first OMIP cycle and 1958–2024 from the second, which gives a total of 130 years (100 samples for training). Considering that up to 2600 virtual Argo profiles appear in a single year within our bounding box around the target latitude only 100 data samples are a proportionate small number for a training process of neural networks. The original VIKING20X experiment contains six OMIP cycles with less AMOC variability in the later cycles (Schiller-Weiss et al., 2025). Although the six OMIP cycles reuse the
atmospheric forcing, the additional cycles provide valuable variability in the ocean state due to the long-term evolution of the AMOC and its link to the density structure and associated thermohaline circulation.

    Extending the training dataset using all available OMIP cycles (Figure 4 Reconstruction (full cycles)) boosts performance on yearly time scales, and hence highlights the benefit of diverse training data for improving the neural network reconstructions. At the 90-day timescale, all four skill metrics show similar performance regardless of training data size, suggesting no
improvement. However, the quantitative performance at the yearly time scale strongly benefits from the additional training data. While the reconstruction for two cycle training and six cycle training mostly agree up to 2012 for the yearly timescale, the six cycle training is able to compensate the former overestimation.

### 4.1.2   Generalization onto the third forcing cycle

A larger amount of training data helps with the yearly reconstruction skill but the variability on test data is limited due to the
20 year period. To show the reconstruction performance on a longer test period we choose the third OMIP cycle in Figure 6

as the test period. Surrounded by 20 years of validation gap on each site. All other parameters of this experiment (see Table 1 Exp.2) are left unchanged compared to the previous. The third cycle shows a smaller AMOC amplitude across its two phases compared to the previously used first OMIP cylce. It has a strong phase during the forcing years from 1987 to 1997, and a weak phase from 2005 to 2015 (Schiller-Weiss et al., 2025), as seen in the ten-year low-pass filtered AMOC of all six OMIP

cycles (Figure 5). While the third cycle is time-wise in the center of the training data this hold also for the AMOC strength in the respective phases. Since the AMOC experiences a long-term decline due to model adjustment, the earlier cycles have strong AMOC amplitudes while the subsequent cycles weaken even more than the third cycle, making it a suitable candidate to test generalization on intermediate conditions. This setup provides a longer test period for evaluating yearly reconstructions and includes both strong and weak-AMOC phases. It even allows for a preliminary assessment of reconstructions at five-yearly

timescales.

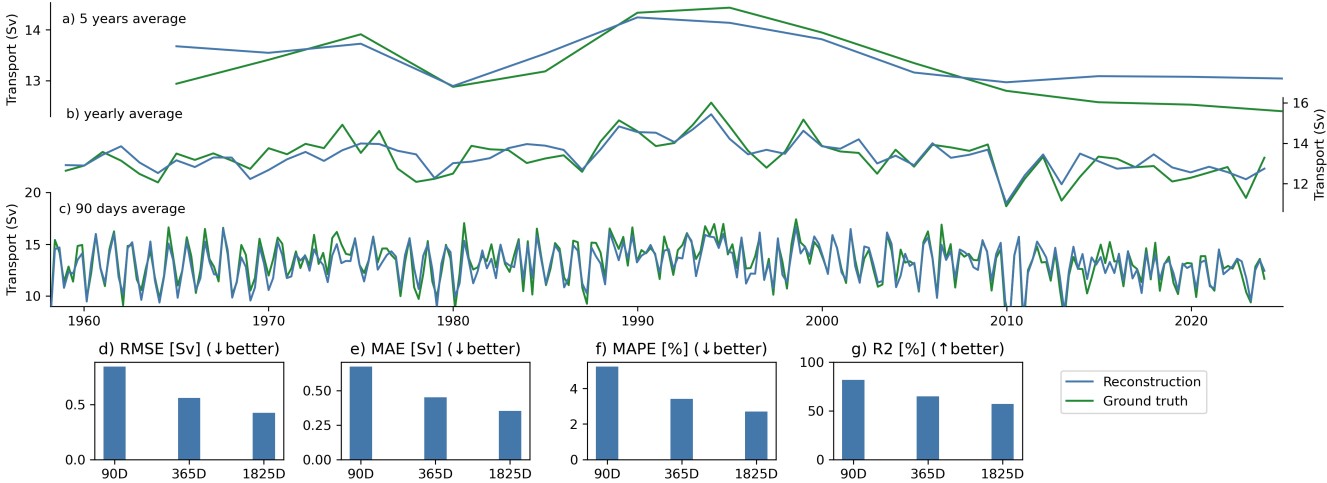

**Figure 6.** Reconstruction (blue) on five-yearly a), yearly b), and seasonal c) time scales with the AMOC of the third OMIP (green) cycle as test period, validation period of 20 years were attached previous and after the test period, the rest of the six cycles was used for the training of the reconstruction.

The framework reconstructs interannual variability in the third cycle with similar accuracy to the shorter test period, capturing both strong-AMOC and weak-AMOC phases. At seasonal time scales (Figure 6 c), the architecture reconstructs the strong interannual signal across the full test period with similar performance (MAPE $5.2\%$; R2 $81.8\%$) as in the two cycle experiment (Figure 4 a)). During the strong-AMOC years (1993–1997), the reconstruction shows lower interannual variability. This sug-

gests that the model does not rely on a default cycle (e.g., surface temperature patterns in Argo) but adapts the amplitude to the specific conditions in that period. With the longer time series, the stronger AMOC phase is included in the test period and sets the dominant variance of the time series compared to the previous test period from 2004 - 2023 in the first cycle. Visually, the reconstruction mostly agrees with the AMOC in the second half of the time series. From 1986 onward it matches the strong peak in 1995, and the low AMOC in 2010. However, on yearly time scales the reconstruction is slightly off in the 1970s and

early 1980s, which was not observed on the seasonal and five yearly time scales. We speculate that this is connected to the lower quality of the wind forcing for the period before the SSM/I-based wind speed product start in 1988, which is used in the JRA forcing (Tsujino et al., 2018) that drives the VIKING20X simulation. Frajka-Williams (2015) showed that the interannual variance of the AMOC signal at the study latitude 26.5°N is driven by the zonal wind stress. The connection between the forcing before SSM/I and the wind stress being the one driver of the interannual variance is our explanation for the misaligned reconstruction before the early 1980s.

As we evaluate a much longer test period we include also the five year time scale into this evaluation. Despite limited training data, the reconstruction shows promising skill on five-year averages (Figure 6 a)). It captures long-term AMOC shifts, although the reconstruction dampens the amplitude and can not reconstruct the full decadal variability. A major issue is the data scarcity but this experiment shows that our reconstruction is likely to work on longer time scales. Although we used all six OMIP cycles, a sufficient number of five-yearly training samples would require additional OMIP cycles or additional ocean simulations, as we later discuss.

## 4.2 Importance of individual components for the AMOC reconstruction

To interpret the reconstruction process, we apply explainable AI (X-AI) techniques (Dwivedi et al., 2023) that reveal how each input contributes to the neural network's estimation of the AMOC. X-AI aims to make typical black-box behavior of neural networks transparent, providing insights into decision-finding process and evaluate the meaningfulness of the inference. The meridional transport at the RAPID array (26.5°N) represents an integrated sum of Ekman transport, transport in the Florida Strait, and upper-mid ocean geostrophic transport (Kanzow et al., 2008). For estimating the AMOC in the real ocean, Ekman transport can be estimated from satellite-based zonal wind stress, and Florida Current is available as a published product. In contrast, the upper-mid ocean transport requires more effort, relying on moorings and repeated ship sections. We hypothesize that Argo profiles have a larger influence on the reconstruction for the longer time series. The main motivation here is that variability beyond the interannual timescales is mainly determined by thermohaline changes which are reflected in the density structure of the open ocean (Biastoch et al., 2008). Experiment 3 (see Table 1) calculates the feature importance (Fisher et al., 2019) for each input component (wind stress, Florida Current, Antilles Current, and Argo profiles) and will provide insight in whether the reconstruction acts on longer time scales as expected. Feature importance quantifies how much each input component contributes to the reconstruction. The fully trained network from Experiment 1 (see Table 1) defines the baseline performance. By removing the influence of an input component, the degradation of skill scores compared to the baseline describes the feature importance. As we always compare the degradation of the total performance, we achieve comparability among the input components. To remove the influence of one input component, we randomly permute its values in the test data for ten times.

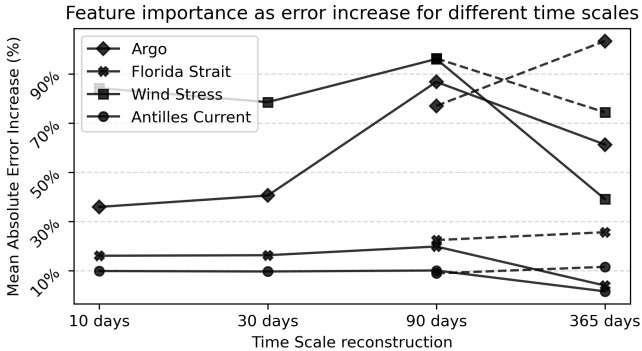

**Figure 7.** Feature importance for each input component based on the MAE ratio between the fully trained reconstruction baseline and evaluation with randomly permuted features. The larger the value is the higher the importance of this input component. The solid lines represent the two OMIP cycle trained reconstruction from Exp.1 (Table 1) and the dashed lines from the reconstruction using all six OMIP cyles.

The feature importance analysis reveals how different inputs influence the AMOC reconstruction across time scales. Figure 7 shows the feature importance with an higher importance being equivalent to a higher increase in the MAE skill when the component is removed from the evaluation. Across all time scales the importance of both currents Florida and Antilles are the lowest and stays constant around a 20% worse MAE skill score. The black solid lines show the experiments with two-cycles from Figure 4 while the dotted black lines represent the six-cycle trained version. Zonal wind stress and the Argo component
show the highest importance with up 90% increase in the MAE. On short time scales the high importance of zonal wind stress is not surprising and agrees with current literature (Moat et al., 2020). However, a high importance of wind stress on shorter time scales partially validates the learned AMOC reconstruction, as it utilizes expected and known correlations.

  Argo becomes more important on longer time scales, while the wind stress importance stays constant or slightly decreases. For reconstructions trained on the first two OMIP cycles of VIKING20X (black lines), the importance of the Argo component
increases up to the seasonal time scales. With both Argo and wind stress importance raising, still the wind stress is more important on time scales up to seasonal for the reconstruction. The turning point is the yearly time scale, which aligns to our hypothesis that on larger time scales, the Argo profiles take the lead from the wind stress for the most important input component. However, the gap between these is marginal for the importance calculated only on two of the OMIP cycles. On the other hand, the feature importance calculated on the six OMIP cycle experiment (Figure 4) shows a more pronounced
increasing trend of the Argo profile importance. In summary, reconstructions at longer time scales rely increasingly on the Argo component shown by an increasing focus of the AMOC reconstruction on interior measurements like Argo profiles.

  In contrast to Argo profiles and wind stress, boundary currents contribute minimally to the AMOC reconstruction, especially on longer time scales. This finding is in-line with recent work about the signal-to-noise ratio (McCarthy et al., 2025), finding only a small signal in the Florida Current transport. However, feature importance only reflects the importance in a fully trained
reconstruction. It does not indicate if an input component can be left out from the input, referred to by fidelity (Markus et al.,

2021). In VIKING20X, the Florida Current transport time series and the AMOC time series for the 90 days time scale have a linear correlation of 0.58 for the first two cycles shown in Figure 1. Removing the influence of the Florida Current transport increases the MAE by only 20%, suggesting that other inputs carry overlapping information. When calculating the feature importance with regard to the interior geostrophic transport, not shown in this manuscript, we observed a similar pattern, the

Florida Current gains importance for seasonal scales, due to the strong correlation to the AMOC (Frajka-Williams et al., 2016). Despite this the importance for all other inputs are constant for the interior geostrophic transport across all time scales with virtual Argo profiles being the most important.

### 4.3 Reconstruction of an RAPID-like interior geostrophic transport within the ocean model

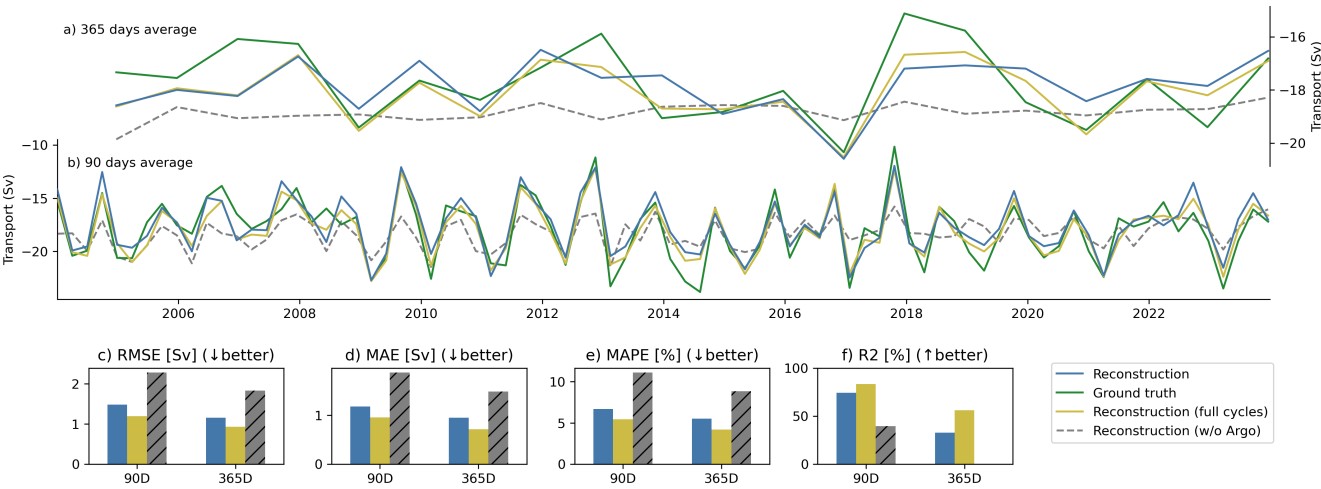

**Figure 8.** AMOC reconstruction trained with the RAPID-like interior geostrophic transport that is calculated from virtual moorings within VIKING20X in yearly a), and seasonal b) time scales. The skill metric of the time scales are shown in RMSE c), MAE d), MAPE e), and coefficient of determination $R^2$ f). The additional gray lines and bars show a reconstruction trained and tested without the Argo profiles but keeping all other input components.

This section investigates whether Argo profiles can help to effectively reconstruct the interior geostrophic transport of the

RAPID upper mid-ocean component. The rising importance of Argo profiles at longer time scales (see Figure 7) suggests a stronger role of the geostrophic interior transport in AMOC variability. To validate our reconstruction further in this experiment (see Table 1 Exp.4), we define an additional target: a geostrophic interior transport time series (see section 2.4), aligned with the RAPID array's mid-ocean transport component. The contribution of RAPID like interior geostrophic transport we find is to be predominantly negative transport. Between our calculated interior geostrophic transport time series and the AMOC we

found a linear correlation of 0.38 on the seasonal timescale showing that the limitation of observations and missing boundary and surface processes change the characteristic of the geostrophic time series. Nonetheless, targeting a reconstruction of this

geostrophic time series will provide insight in the suitability of Argo floats to be a proxy for the interior transport as it is measured by arrays like RAPID.

We now evaluate how well the reconstruction framework can capture the geostrophic component of the AMOC. The geostrophic reconstruction uses all input components to utilize all possible data correlations, which is guided by the fact that the different components are not completely independent. For example, it was found by Frajka-Williams et al. (2016) that the Ekman signal contains information about the deeper density structure and the western boundary current. The reconstruction of the interior geostrophic transport (see Figure 8) performs best at seasonal time scales, revealing differences in predictability between AMOC components. On time scales shorter than the seasonal scale, the reconstruction is not comparable to the performance of the AMOC which is expected because the variability of the AMOC is mainly wind-driven (Kanzow et al., 2010). Hence, we not include those shorter time scales in the evaluation.

At seasonal resolution (see Figure 8 b)), the reconstruction captures prominent interannual features of the interior geostrophic transport, suggesting a physically meaningful signal. The time periods with strongly reduced interannual variability in 2013, and 2022 are well reconstructed, suggesting that our reconstruction framework does not compute a static seasonal signal but rather adapts to the dynamical regimes. Although we do not use any absolute timing feature as input (e.g. year, month, or day) from which such a periodical signal could be trivially inferred. However, we believe that information about the phase of the seasonal cycle is still present in the hydrographic data and hence, could still be inferred by the reconstruction. For example, Argo profiles include mixed-layer measurements, which in the subtropical North Atlantic are sufficient to infer the seasonal cycle from surface-near layers. Mostly the seasonal reconstruction struggles with the strong southward events as in 2013, and 2014 by underestimating the amplitude.

Despite improvements at seasonal scales, the reconstruction of the interior geostrophic transport loses accuracy at yearly resolution, pointing to limits in data coverage and training diversity. With the yearly time scale we see a similar drop from seasonal to yearly in the explained variance for the two cycle OMIP training of the AMOC (Figure 4 g). Interestingly, the reconstruction trained on all OMIP cycles with the geostrophic target variable drops to 50% of explained variance (Figure 8 f)) which in contrast kept stable for the AMOC. The additional cycles from the two cycle training to the six cycle training in the geostrophic reconstruction increase the performance on the seasonal time scale across all metrics. This improvement was not observed with the AMOC, indicating that more training data benefits the geostrophic time series reconstruction similar to the more training data for the yearly reconstruction of the AMOC. This opens the question whether the yearly reconstruction of the interior geostrophic transport can still be improved by more heterogeneous ocean states in the training data. One option is to reuse the same training samples but using different spatial Argo distributions to augment the existing data; another is adding additional ocean simulations to the reconstructions. However, both approaches begin to address the broader problem of transfer learning for applying reconstructions to real-world data, which we outline in the discussion but lies beyond the scope of this study.

Important for our study is the comparison of the interior geostrophic transport reconstruction with the Argo profiles as described before and a reconstruction trained without any Argo profiles or information about the deep ocean. Such a reconstruction that only utilized zonal wind stress, Florida Current transport, and Antilles Current transport is shown in the gray dashed

lines in Figure 8 a,b) and as gray bar in the corresponding skill plots c-f). We retrained the reconstruction in a neural network without argo-related inputs. This removes the influences of the Argo profiles completely from the reconstruction. Across all skill score a strong performance drop is present between the reconstruction with and without Argo profiles, highlighting the importance of interior measurements for our reconstruction. While the seasonal comparison shows that the reconstruction without Argo profiles underestimates the amplitude of the seasonal variability, the reconstruction on the yearly scales can hardly reconstruct any interannual variability but rather predicts a constant mean function. Despite the limited training data for the interior geostrophic transport at yearly time scales the comparison to an reconstruction without any Argo floats pinpoints the importance of the Argo component in our reconstruction.

## 4.4 Importance of the deep geostrophy

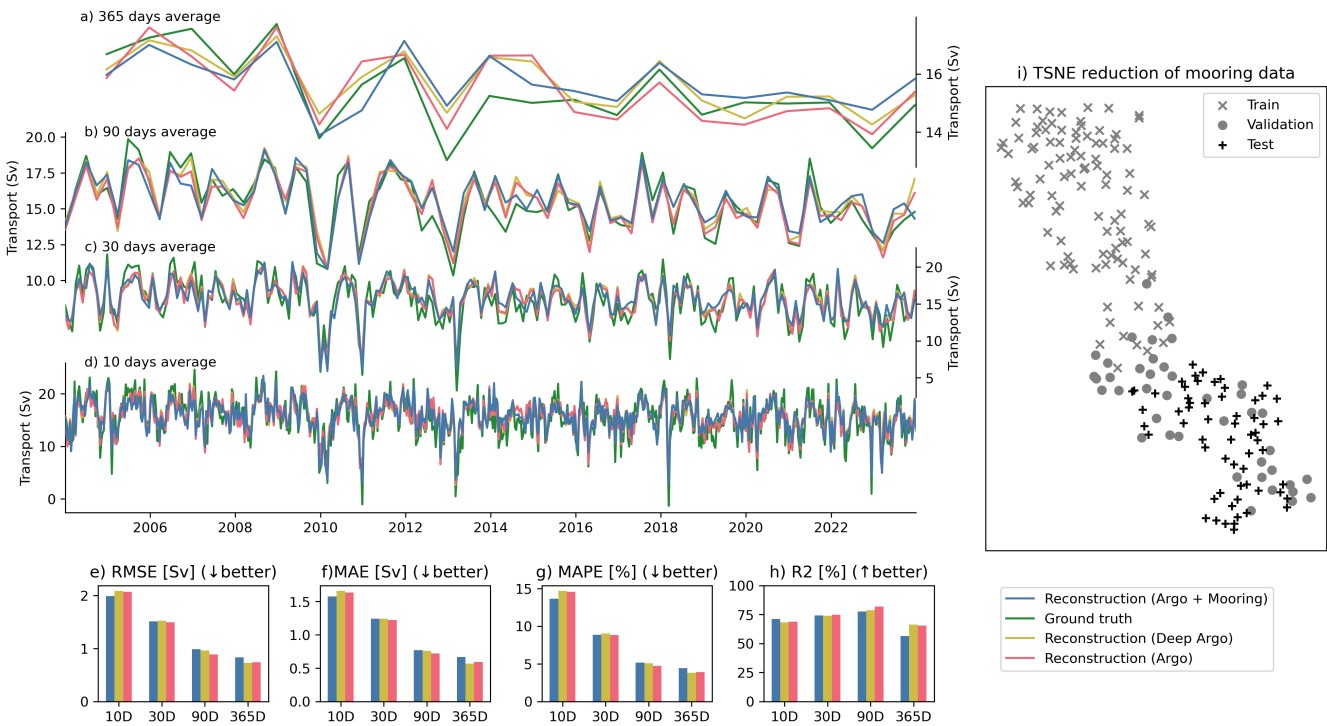

**Figure 9.** Reconstructing the VIKING20X AMOC on yearly a), seasonal b), monthly c), and ten-days d) time scales with Argo reaching 2000 meter depth (red), Deep Argo floats reaching up to 6000 meter depth (yellow) and Argo floats up to 2000 meters as well as the deeper mooring information (blue). The skill is shown in RMSE e), MAE f), MAPE g), and coefficient of determination $R^2$ h). For a deeper insight into the performance we show the TSNE reduction of the ocean state estimation (Figure 2 B)) for the seasonal time scales and the mooring version split into training, validation, and test data.

The standard Argo network, which samples the ocean only down to 2000m, may not capture key features of deep ocean variability which could influence AMOC reconstructions. Thus, the reconstruction cannot rely on information from layers

below 2000m, raising the question of how essential deeper data is for accurate reconstruction. In the previous experiments, the reconstruction was supplemented with the virtual moorings between the bottom and 2000m depth. To assess the potential value of deep-ocean information, we compare in this experiment (see Table 1 Exp. 5) reconstructions using standard Argo profiles, exchanging all Argo profiles with Deep Argo profiles (to 6000m), and standard Argo profiles supplemented with mooring-derived transport below 2000m. In the Deep Argo experiment, we assume a hypothetical scenario in which Deep Argo floats offer the same spatial coverage as as the standard Argo array.

Across all timescales, we found that the added value is limited by the seen training data, which did not cover heterogeneous ocean states in the deeper layers, leaving the possible influence of Deep Argo floats undetermined in our training setup. The similar patterns in the reconstruction suggest that information below 2000m may have limited influence on the reconstruction performance, even though minor differences appear in the skill metrics (MAE, RMSE, and MAPE). The virtual standard Argo profiles combined with the virtual moorings perform best at the 10-days time scale, but show the worst performance at longer time scales. This is even more highlighted in the R2 score in Figure 9 h), which shows the expected decline in performance at yearly scales. Interestingly, the decay is the strongest for the combined Argo and mooring data.

To understand the performance drop in the mooring-supported reconstruction, we use t-distributed Stochastic Neighbor Embedding (t-SNE (van der Maaten and Hinton, 2008)) to assess whether deep ocean states in the test set are represented in the training data. The algorithm is part of the manifold learning family and maps points from a high-dimensional space into lower dimensions visualization purposes. Using t-SNE, we visualize clusters of the virtual moorings (below 2000 meter) across the training, validation, as test data. As shown in Figure 9 i), the test data is almost completely separated from the training data, indicating a significant distribution shift between them.

These findings suggest that while deeper observations can improve reconstructions, their benefit depends on having sufficient representation in the training data. Although not shown in this experiment, we do expect the deeper observation to contain useful information about the stream function at depth. The performance drop occurs because, during the test period, the reconstruction is forced to extrapolate in unseen data regions. To better leverage mooring data, the training must include more heterogeneous deep ocean states that are representative for the test period.

## 5   Discussion

In this study, we presented a framework that is based on neural networks to reconstruct the Atlantic Meridional Overturning Circulation (AMOC) across time scales ranging from 10-days to yearly and longer using observation-like input data. We trained the framework on consecutive forcing cycles produced under the OMIP protocol within the eddy-rich VIKING20X ocean model, focusing on both the AMOC and the geostrophic component. The latter mimics the interior transport from the AMOC, estimated based on the RAPID-array moorings. Our results show that a) reconstruction skill varies significantly with time scale and input type, b) Argo profiles are a promising replacement option for the reconstruction of the RAPID-like interior geostrophic transport, and c) the amount and heterogeneity of the training data is important especially for accurately capturing the longer time scale variability. A core strength of the framework is its use of a graph neural network that is tailored to

sparse, and irregular distributions of Argo floats, enabling the model to learn spatio-temporal patterns from realistic Argo float distributions without the need for interpolation or binning to spatially regular structures. These findings underline the potential of data-driven reconstructions with observation-based measurements for variability estimates of the AMOC and highlight the data limitations that must be addressed to enable reliable long-term AMOC monitoring with Argo floats.

In accordance with previous studies (Solodoch et al., 2023), we also found a strong relation between the zonal wind stress and the AMOC on short time scale of up to 30 days. However, we can not reproduce the results of over 90% explained variance seen by Solodoch et al. (2023). This might have different reasons. We have seen that the AMOC undergoes substantial variability on pentadal and decadal time scales which can be understood as a changing background state of the AMOC. Testing a reconstruction on previously unseen decades of data requires generalization of the reconstruction to different decadal

background states, a challenge which is not present for the 5-year testing period available to Solodoch et al. (2023) and the absence of a sufficient validation period that isolates training from testing data. Our 20-year testing period padded with 5 years of validation data requires our reconstruction to handle the underlying decadal variability. Hence, it is not surprising that our explained variance does not meet the very high values seen in the previous study. Due the length of our test period and the heterogeneous states of the AMOC which are covered in each of the OMIP cycles we can not apply a linear detrending

algorithm to the transport series as previous reconstruction studies did (Sanchez-Franks et al., 2021; Solodoch et al., 2023). This highlights that for achieving a reconstruction that potentially generalizes to unseen dynamical regimes, the training and test data need to be chosen and preprocessed carefully.

On longer time scales, we identified an increasing importance of the Argo profiles which we used as a widely available observation product to monitor the upper cell of the Northern Atlantic Deep Water at the RAPID latitude. While other studies

use ocean bottom pressure (OBP) for the estimation of the interior ocean transport (Solodoch et al., 2023), we highlight the large effort for calibration and the instrument drift (Herrford et al., 2021). Satellite-based OBP estimates from e.g., the GRACE mission (Landerer et al., 2020), require a high spatial resolution for a good reconstruction at the continental shelf (Delman and Landerer, 2022). Hence, satellite-based OBP estimates may be promising for AMOC reconstruction in the far future when decades of recorded data exists from satellites yet to be deployed. The advantage of Argo floats over satellite-based OBP is

their longer availability since the early 2000s. The advantage of Argo floats over mooring-based estimates is their presence in regions currently not covered by moorings. Hence, and although other regions may have lower coverage by Argo floats, there is potential to train an AMOC reconstruction at other latitudes. Using Argo profiles showed a great performance to approximate the geostrophic component of the AMOC with the major limitation being the amount of available training data. We have shown that all six OMIP cycles have to be used in the training for a reconstruction on yearly time scales. Interestingly,

the additional later OMIP cycles (3-6) increased the reconstruction performance for both the AMOC and interior geostrophic transport. Although, the strength and variability of the later cycles is lower compared to the test period in the first cycle. Such an improvement shows that additional training data can be acquired from data that represent other ocean states, making a case for training with a model ensemble for potential further improvement. However, data from different OMIP cycles do not represent linearly independent samples due to the repeated atmospheric forcing. Hence, using over 300 years of data from all OMIP

cycles is not comparable to a 300-year coupled ocean atmosphere simulation. A particular advantage here is the long-term,

model-related, drift of the AMOC in VIKING20X (Schiller-Weiss et al., 2025). We utilized this to verify the performance of the Argo profiles at the complete third OMIP cycle (Exp. 2 Table 1). It showed a well-reconstructed annual-mean AMOC and highlighted that our approach is applicable in other transient dynamical states of the AMOC.

We isolated in this study the interior transport in a RAPID-like fashion from the ocean model output. Therefore we utilized virtual moorings to best approximate the method of the RAPID upper mid-ocean transport component with it coverage rate and assumptions. We focused on the geostrophic part of the upper mid-ocean RAPID component to investigate to which extent Argo profiles can be helpful in reconstructing the deeper southward return flow of the AMOC that is based on geostrophic transport (Elipot et al., 2014). For longer time scales, we found a good skill of reconstructing the geostrophic part of the AMOC from Argo floats. This is a promising finding for possible later transfer of our approach to real-world observations. From the RAPID array, the mooring array covering the interior transport is the perfect test bed for assessing to which extent neural networks can estimate the interior transport based on Argo profiles. We delivered a first proof of concept by calculating a RAPID-like interior transport and achieving a well-reconstructed time series at 1000m depth based on virtual Argo data.

For a routine application on the RAPID array and other arrays, this study also identified limitations. The training within VIKING20X with virtual Argo profiles has shown that an AMOC reconstruction on interannual or longer time scales requires large amounts of diverse training data. This limits a naive application of our framework to real Argo floats due to only 20 years of available data with a constant coverage of Argo profiles only after 2012. However, for shorter timescales and use cases like the filling of smaller temporal gaps the observational data within the next decade could be sufficient to train an AMOC reconstruction purely on real-world data. A promising approach for longer time scales are transfer-learning methods, which allow neural networks to be pre-trained on larger similar datasets, e.g. from model simulations, that are then transferred and executed on scarce target-domain data, e.g. real RAPID observations. Such a transfer performs best, the more similar the data of the training and the targeted domain are. Due to model biases it is not guaranteed that ocean simulations have realistic spatial distribution of heat and salt, although the variability of the large scale measurements like AMOC appear to match. A further reduction of those model biases and an even higher spatial and temporal resolution will certainty be helpful for a transfer albeit possibly not feasible in the near future. Longer training periods are already seen in literature for the reconstruction of AMOC with neural networks as for example 8000 annual means from the Community Earth System Model presented by (Wu et al., 2025). However, we argue that for a future application to the real-world ocean, longer simulations of high-resolution ocean models will be necessary to minimize the bias.

Nonetheless, our study points out that Argo profiles can be a promising alternative for the reconstruction of interior transports together with the finding that Deep Argo floats and mooring data below 2000m can not yet be used for our reconstruction approach. This is contrary to the large signal to noise (SNR) ratio that McCarthy et al. (2025) found between the Lower North Atlantic Deep Water and the AMOC. But the SNR based on a linear regression is calculated with five year filters, suggesting that the deep geostrophy will play a role on timescales longer than those investigated in this study.

The reconstruction of the AMOC time series from related observational instruments (e.g. satellite-based altimeters for measuring sea-surface height) is of large interest to monitor its variability at other latitudes or reduce the cost for measurement arrays like RAPID, or OSNAP. Such reconstructions can be based on various methods ranging from purely statistical (Willis,

2010; Frajka-Williams, 2015; Sanchez-Franks et al., 2021) to very powerful AI methods (Solodoch et al., 2023; Wu et al., 2025). Our study fits with this theme showing the potential and opening an interesting direction of research with basin-wide Argo profiles and advanced AI methods to reconstruct the AMOC strength in a high resolution ocean model. However, as mentioned, the transfer from the ocean model to a reconstruction of the real-world RAPID measurements will require further investigations. Here, we set the stage for these further investigations by reconstructing a RAPID-like interior geostrophic transport and by elaborating on the importance of training data sizes and the potential leakage from training to the evaluation. This work emphasizes the chance for model-data fusion using advanced AI methods to better monitor the real-world AMOC at different latitudes of the Atlantic Ocean.

*Code and data availability.* The artifacts of this study are divided into two datasets and two software artifacts. The study is based on an isolated software framework(Wölker, 2025a) to ensure further development of the approach. For the reproducibility of our results and the approach we published the input components and target variables that we extracted from the ocean simulation in this dataset (Wölker et al., 2025). To rerun the experiments or recreate the figures of this study we published another software repository (Wölker, 2025b). In addition, we published the experimental results for this study which are the base for the figure creation (Wölker, 2025c). The Argo data were collected and made freely available by the International Argo Program and the national programs that contribute to it. (https://argo.ucsd.edu, https://www.ocean-ops.org https://doi.org/10.17882/42182#110199). The Argo Program is part of the Global Ocean Observing System. (Argo, 2025)

## Appendix A: RAPID-like interior geostrophic transport calculation

To create a interior geostrophic transport time series, comparable to the interior transport estimates from the RAPID array, we implemented our own calculation from virtual moorings in the VIKING20X ocean model. Here, we briefly describe our assumptions and the calculation from the daily-mean ocean model output. The overall process is based on the description from McCarthy et al. (2015) and has two main steps, i) the extraction and merging of mooring-based density profiles, ii) calculation of the basin-wide transport. All steps are also part of the published software code (Wölker, 2025a).

### Extraction of virtual mooring density profiles

For a given mooring location we identify the closest location on the $1/20°$ model, calculate in-situ density for each z-level and linearly interpolate the resulting profiles to 20 meter z-levels, corresponding to the published RAPID mooring data. We extract virtual mooring density profiles for the following RAPID mooring codes: WB3, WBH2, WB2, MARWEST, MAREAST, EB1, EBH1, EBH2, EBH3, EBH4.

We group the virtual in-situ-density profiles into four groups representing the western and eastern boundary and the western and eastern flank of the Mid-Atlantic Ridge. Then we merge the profiles in such a way that for each z-level, the resulting profiles always contain the density estimate that is closest to topography. This results in four virtual in-situ-density mooring profiles, the two boundary moorings $(\rho_W, \rho_E)$, and the two moorings $(\rho_{MW}, \rho_{ME})$ at either side of the Mid Atlantic Ridge.

### Calculation of basin-wide geostrophic transport

The goal is to estimate a transport time series for the $1000m$ z-level. To this end, we first calculate vertical shear for three compartments of the zonal section. Two deeper compartments span the waters between the ocean floor and $3700m$ depth west and east of the Mid-Atlantic Ridge. The third compartment spans the water body between the eastern and western boundary between $3700m$ depth and the surface. As the real RAPID mooring profiles do not cover the water column all the way up to the surface, and as we want to mimic the data coverage of the real RAPID moorings as closely as possible, we mask out virtual-mooring data shallower than a certain depth $z_{mask}$ which is different for each real RAPID mooring profile we are mimicking. Therefore, we sample uniformly from the corresponding RAPID mooring the depth shallowest instrument depth, similarly as we sample also the position of the virtual Argo profiles. We compute the zonal mean of vertical shear of northward velocity for all z-levels $z_i$ deeper than $z_{mask}$ from the virtual density profiles and assume constant shear for depths shallower than $z_{mask}$.

$$\frac{\partial v}{\partial z_i} = \begin{cases} -\frac{g}{\rho_0 f} \frac{\rho^W_{z=z_{mask}} - \rho^E_{z=z_{mask}}}{x^W - x^E} & \text{for} \quad z_i > z_{mask} \\ -\frac{g}{\rho_0 f} \frac{\rho^W_{z_i} - \rho^E_{z_i}}{x^W - x^E} & \text{for} \quad z_{mask} \geq z_i \geq -3700m \\ -\frac{g}{\rho_0 f} \left( \frac{\rho^W_{z_i} - \rho^{MW}_{z_i}}{x^W - x^{MW}} + \frac{\rho^{ME}_{z_i} - \rho^E_{z_i}}{x^{ME} - x^E} \right) & \text{for} \quad -3700m \geq z_i \end{cases} \tag{A1}$$

For later use, we define the width of the water body at depth $z$ as $L_x(z) = x^W - x^E$ for the upper compartment and $L_x(z) = x^W - x^{MW} + x^{ME} - x^E$ for the lower compartment. From, this, we can estimate zonal mean northward velocity at depth $z$ referenced to the surface velocity as

$$v(z)|_{0m} = \int_z^{0m} \frac{\partial v}{\partial z_i} \, dz' \tag{A2}$$

and northward transport relative to the surface as

$$V(z)|_{0m} = \int_z^{0m} L_x(z') v(z')|_{0m} \, dz' \tag{A3}$$

Following McCarthy et al. (2015) we choose a reference level of no transport of $4820m$ marking the bottom layer of RAPID measurements and not including the overturning of the Antarctic Bottom Water. Then we calculate the interior geostrophic transport $IGT$ contributing to the $1000m$ overturning as

$$IGT = V(-1000m)|_{-4820m} = V(-1000m)|_{0m} - V(-4820m)|_{0m} \tag{A4}$$

Selecting the $1000m$ overturning avoids the detection Antarctic Intermediate Water (AAIW) as done by (McCarthy et al., 2015). However, in a direct comparison to the Rapid timeseries two reconstructions can be trained one for 1000 meter and another for 700 meter. We argue that such an extraction following RAPID as close as possible is necessary to create an accurate time series that contains all the characteristics and caveats of the observations and to train the AMOC reconstruction most realistically to find reasonable correlations. A similar computed AMOC series is a strong prerequisite for any assumptions about transferability between the ocean model VIKING20X and the real-world observations within RAPID.

*Author contributions.* AB: initiated the idea, conceptualized the approach, analysis of the results; MR: initiated the idea, design of the machine learning method; WR: contributed to design of technical concepts, improving the validity of the approach, analysis of the results; YW: implemented and performed the experiments, design and implementation of the approach and machine learning method, analysis of the results, written the first draft - All authors reviewed and improved the draft over iterations.

*Competing interests.* The authors declare that they have no conflict of interest.

*Acknowledgements.* The first author Yannick Wölker is funded through the Helmholtz School for Marine Data Science (MarDATA). The study was supported by the European Union's Horizon 2020 research and innovation program under Grant Agreement 818123 (iAtlantic). The authors gratefully acknowledge the Earth System Modelling Project (ESM) for funding this work by providing computing time on the ESM partition of the supercomputer JUWELS at the Juelich Supercomputing center (JSC). We thank the two reviewers of this study for their detailed comments and suggestions.

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
