# Peer review of "Estimating the AMOC from Argo Profiles with Machine Learning Trained on Ocean Simulations"

_EGUsphere, 2025_

## Referee Comment (RC1)

Review of **egusphere-2025-2782** "Estimating the AMOC from Argo Profiles with Machine Learning Trained on Ocean Simulations" by Yannick Wölker , Willi Rath , Matthias Renz , and Arne Biastoch

My expertise as a reviewer is mainly focused on the data science part, the parameter estimation, machine learning, statistics, error handling, and significance considerations, in the frame of oceanographic research questions.

General:

- The manuscript represents an important and interesting study on the potential of Argo floats to be used to estimate the AMOC, specifically the geostrophic part.
- I recommend publication after some minor clarifications, improvements and corrections.
- You mention the challenges of handling the irregular Argo data, which is reasonable. Then you overcome these difficulties with the embedding and graph-based NN approach, which is technically a very smart solution. However, you did not try an interpolation approach, bringing the Argo data on a regular grid and using those data as input for the feedforward NN. Thus, we don't know if your graph-based approach is superior. In the end your results are good, which probably justifies your approach, however, for me it's always the question if these results could have been achieved with simpler methods?
- Your approach, based on model data, shows that there is potential to reconstruct the AMOC utilizing Argo floats. However, for an application with real data, not enough data is available for a NN approach. So, what is not fully clear to me from the manuscript is, when we can reach "enough data"? Or, regarding to your discussions, is the only solution transfer learning, and enough data will not be available in a reasonable near future? Can you please clarify that?
- It is important to thoroughly always differentiate between the real elements in this study (AMOC, Argo, etc.) and the simulated. Please check all text.

Specific:

- Line 9-10: Add "… AMOC can be **potentially** data-drivenly …"
- Line 21: Are you referring to the North Atlantic Deep Water? But that is colder and **saltier** not fresher, or?
- Line 27: You say that ocean and climate models often fail to simulate the AMOC, but nevertheless you go for a full model analysis to draw inference on the real world.
- Line 53: Again make clear that you are not using real Argo.
- Line 93-95: I understand that the authors are going for a NN approach and real data is too limited in this case. However, why do the authors think that simpler approaches like linear regression may not be sufficient?
- Line 110: This sentence is confusing, what do you mean by "widely available observations"? If I understood correctly you are not using real observations.
- Line 113: You are not using Argo data!
- Line 110-124: Start the whole paragraph with explaining that you use simulated data.
- Page 5, Figure 1: c,d,e: Shouldn't the middle x-label be the other way round 2024/1958?
- Line 182: First Argo floats have been deployed since 1997.

- Line 284: Difficult to understand. What is the "trained reconstruction"? And what means "the trained reconstruction is able to reconstruct…"?
- Line 303-327: You are saying that if you neglect the spatial info on Argo data, you can utilize a suitable neural network architecture. In the following you say you keep the spatial component using SUSTeR. I do not understand what in the end you do. In addition, understanding SUSTeR and explanations about traffic are not helpful. I suggest to remove this explanation and refer to the publication. Instead please make clearer what you have actually done in the end.
- Line 329-- Sect. 3.2: Regarding the training procedure of a NN, it would be interesting to see a "loss curve". Often these loss curves are given for the performance of the model on the training set, during the training, as well as on the validation set (unknown).
- Line 374, Eqn. 4: I think the denominator is not Var(y), but the total sum of squares \sum{ (y_i-\overline{y})² }.
- Page 16, Fig. 4: I suggest to plot the reconstruction curve (blueish) on top of the ground truth (green) to better see it.
- Line 595: I guess you again randomized the Argo input data, not leaving it really out? Please mention in the text.
- Line 619: If I understand correctly by reading the full paragraph, the reason for no added value of deep Argo is probably just caused by not having enough training data. Thus the influence of deep Argo stays rather unknown. If that is true, please mention already here.
-

Corrections:

- Line 41: Rewrite this part, which sounds strange "… cables that measurement …"
- Line 48: Space missing "… balance(Mc …"
- Line 155: Figure ??
- Line 259: "… an high …" → " … a high …" and "… an dedicated …" → "… a dedicated …."
- Line 309: (?
- Line 396: "… due **to** larger …"
- Line 399: "The **the** …"
- Line 400: "… limits the compare …" → sounds strange, please rewrite.
- Line 427: "data(Jiang" → space
- Line 478: "… and the also the …"
- Line 536: "brach" ? → branch
- Line 542: "… due **to** the ..."
- Line 543: "… transport(Mc" → space
- Line 630: Change "We test if the test data lays within the training data and its ..." to "We investigate if the test data lie within the training data and if its ..."
- Line 706: "… amounts **of** diverse …", delete: "… set of …"
- Line 736: "… mentioned,the …" → space

---

## Author Comment (AC1)

**Final Author comment for RC1**

Review of egusphere-2025-2782 "Estimating the AMOC from Argo Profiles with Machine Learning Trained on Ocean Simulations" by Yannick Wölker , Willi Rath , Matthias Renz , and Arne Biastoch

My expertise as a reviewer is mainly focused on the data science part, the parameter estimation, machine learning, statistics, error handling, and significance considerations, in the frame of oceanographic research questions.

Response: We thank the reviewer for their thoughtful and constructive comments on our manuscript. We very much appreciate the reviewer's expertise in data science and machine learning within the context of oceanographic applications, which has helped us to clarify and strengthen the explanatory aspects of our study. We have revised the manuscript accordingly, as detailed below, and believe these changes have substantially improved the clarity of the paper.

**General:**

- The manuscript represents an important and interesting study on the potential of Argo floats to be used to estimate the AMOC, specifically the geostrophic part.
- I recommend publication after some minor clarifications, improvements and corrections.
- You mention the challenges of handling the irregular Argo data, which is reasonable. Then you overcome these difficulties with the embedding and graph-based NN approach, which is technically a very smart solution. However, you did not try an interpolation approach, bringing the Argo data on a regular grid and using those data as input for the feedforward NN. Thus, we don't know if your graph-based approach is superior. In the end your results are good, which probably justifies your approach, however, for me it's always the question if these results could have been achieved with simpler methods?
  - Response: We thank the reviewer for this interesting comment. We tested their mentioned method in an earlier development phase and quickly moved on due to the following reasons. First, the classical binning comes with a bunch of assumptions about the spatial shape, temporal distance, and connectivity of binning cells. In the process of defining these, we found ourselves in a difficult situation, which would require a lot of testing. However, the results may be valid only in this particular study region. Using the graph-based approach with the learned clustering solves this in a pure data-driven fashion, which we envision to be more practical for future use and applicable for other tasks. Second, a feedforward NN (FFN) on top of a static binning was performing worse because the amount of Argos per bin is variable. As FFN have

a static and global connection, similar to the reason why Convolution Neural Networks are used in image processing, spatial or temporal shifts of correlations require explicit learning in an FFN, while this is inherent in GNNs for irregular topologies and CNNs for regular topologies. Overall, we think that a too deep investigation of the method is beyond the scope of this study, with our main goal showing the applicability of a method that is mostly data-driven and can therefore be reused in different scenarios. Based on the feedback of the reviewer also in the `Specific` section, we restructured the second part of the 'Processing of Argo profiles' section. We added the following text in line 246 to motivate our design choices more. "A static clustering on all data points, could create such a structured input. Around our targeted latitude this would amount to a zonal binning (Willis, 2010; Hernández-Guerra et al., 2010) but the set of Argo floats, especially for the shorter target time scales, show a heterogeneous distribution that would require carefully hand-crafted cluster boundaries. A classical binning comes with even more assumptions about shape, distance, and connectivity of the bins, which would require a thorough testing of hyper parameters. However, we aim in this work for a data-driven mapping function which identifies a structured numerical representation (n-dimensional vector)"

- Your approach, based on model data, shows that there is potential to reconstruct the AMOC utilizing Argo floats. However, for an application with real data, not enough data is available for a NN approach. So, what is not fully clear to me from the manuscript is, when we can reach "enough data"? Or, regarding to your discussions, is the only solution transfer learning, and enough data will not be available in a reasonable near future? Can you please clarify that?
  - Response: We appreciate the reviewer's question as it is an important point. In general, the quality of training data is determined by how well it covers the expected states during inference. This means we would potentially reach "enough" training data when sufficient heterogeneous ocean states were observed, from which general knowledge can be extracted that is most likely to match the ocean state during inference. However, this statement is vague because the question of how the AMOC and its driver change in the future is under heavy discussion. The "enough data" also depends on the targeted time scale. While on short time scales, the current observational data contains different realisations of more frequent signals like the seasonal cycles, this is not true for yearly or decadal signals, which we would be most interested in by using Argo profiles. Based on this consideration, we found simulations to be a promising testbed for the question "What if we had plenty of observed years?" to test our reconstruction. For long time scales, we see the potential in the transfer learning approach with pre-trained reconstruction on large ensembles to cover more heterogeneous ocean states and a fine-tuning phase, much like the hyped foundation models. Considering our demonstrated performance, we see on shorter time scales the potential for training a hybrid reconstruction with assimilated simulation and Copernicus data. In the next decade, enough Argo measurements could be reached to train a stable reconstruction that would assist in scenarios where moored arrays like RAPID may have data gaps. We added to the paragraph with about the applicability and transfer learning methods a distinction between long and short timescales (I.583) "For a routine application \[...\] this study also identified limitations. The training within VIKING20X with virtual Argo profiles has

shown that an AMOC reconstruction on interannual or longer time scales requires large amounts \[...\]However, for shorter timescales and use cases like the filling of smaller temporal gaps the observational data within the next decade could be sufficient to train an AMOC reconstruction purely on real-world data."

- It is important to thoroughly always differentiate between the real elements in this study (AMOC, Argo, etc.) and the simulated. Please check all text.
  - Response: We thank the reviewer for the valuable feedback and agree that a sharp line between the observational and simulated data is essential. We changed the text in the introduction to make a clear statement, that this study is inspired by real-world observation strategies but uses only simulated ocean data for the reconstruction. (I. 94) "In this work, we demonstrate how and to what extent the AMOC can be reconstructed in an ocean model setting from simulated measurements that mimic widely available observational products using supervised machine learning. All reconstruction inputs are extracted from the ocean simulation together with virtual Argo floats which have the same spatio-temporal distribution as in the real-world, and then tested against the total AMOC calculated on the same simulation.". Additionally, we made sure to be consistent with the wording "virtual Argo profiles" whenever we refer to Argo profiles extracted from a simulation.

**Specific:**

- Line 9-10: Add "... AMOC can be potentially data-drivenly ..."
  - Response: We thank the reviewer for their comment and changed the abstract accordingly to "...AMOC can be potentially reconstructed by Argo profiles in a data-driven fashion"
- Line 21: Are you referring to the North Atlantic Deep Water? But that is colder and saltier not fresher, or?
  - Response: Here we refer to NADW which is typically fresher (about 35 psu) compared to the Gulf Stream (36-37 psu) flowing above.
- Line 27: You say that ocean and climate models often fail to simulate the AMOC, but nevertheless you go for a full model analysis to draw inference on the real world.
  - Response: We thank the reviewer for this valuable feedback. Our goal is to mention the deficits of models to point out the extra work that would be necessary for an inference in the real world. The goal of this work is to use models as a physically consistent testbed for the AMOC reconstruction. The bias of the AMOC in an ocean model comes from the global balance which is out of scope for our AMOC reconstruction. The other benefit to ocean simulations is the longer data horizon than the real world. The AMOC biases in the ocean models are introduced by global balances. We changed the text to make clear that, despite the inherent errors of ocean models, these are a valid way to test our reconstruction assumptions.
- Line 53: Again make clear that you are not using real Argo.

- Response: We understand that we have to be clearer in the distinction between real Argo measurements and the "virtual Argo profiles", which we used from ocean simulations. We change the sentence and added the word concept, highlighting that the idea is motivated from real Argo profiles. Also, we specified that we use "simulated Argo floats from ocean models".
- Line 93-95: I understand that the authors are going for a NN approach and real data is too limited in this case. However, why do the authors think that simpler approaches like linear regression may not be sufficient?
  - Response: We thank the reviewer for this important comment. In our view, simpler approaches would require structured data, which does not hold for the spatial and temporal distribution of Argo profiles. Similar to our answer the reviewer's third general comment. We refer to this observation in the previous paragraph (I. 71) "So far, these approaches rely on spatial and temporal binning \[...\] Most machine-learning approaches require structured input data \[...\] ". Overall, the non-linear mechanism essentially allows for learning implicit data imputations, which would need to rely on subjective choices in linear approach and the binning of Argo profiles. The addition of the linear approach would not be possible in an objective sense (because of all the required choices) and would substantially expand the manuscript in length and scientific content beyond the current scope.
- Line 110: This sentence is confusing, what do you mean by "widely available observations"? If I understood correctly you are not using real observations.
  - Response: We acknowledge the unclear statement and thank the reviewer for their finding. The text was removed in a process of removing redundancy for the sake of preciseness. The current paragraph is: (I. 94) "In this work, we demonstrate how and to what extent the AMOC can be reconstructed in an ocean model setting from simulated measurements that mimic widely available observational products.% using supervised machine learning. Reconstructing the AMOC on different time scales from 10 days up to five years, we find that the importance of geostrophic transport in the ocean interior becomes more pronounced, with longer time scales. To capture this, we use virtual Argo profiles as our observational input, leveraging their insight into the ocean interior while addressing their sparse and irregular sampling with a graph-based neural network approach."
- Line 113: You are not using Argo data!
- Line 110-124: Start the whole paragraph with explaining that you use simulated data.
  - Response: We agree with the reviewer that this paragraph requires a more transparent communication of the used values. We moved the sentence "All reconstruction inputs are extracted from the ocean simulation..." from the next paragraph to the top of this one. Also, we made sure to refer to virtual Argo profiles for our reconstruction, because these are extracted from the ocean simulation with spatial and temporal distribution from real Argos.
- Page 5, Figure 1: c,d,e: Shouldn't the middle x-label be the other way round 2024/1958?
  - Response: We agree with the reviewer and changed the x-axis for the x-ticks in the middle to '2024/1958'.

- Line 182: First Argo floats have been deployed since 1997.
  - Response: We thank the reviewer for pointing out our inaccuracy with the start of the Argo deployment. Based on our cited literature Riser et al. 2016 we changed the first deployment year to 1999.
- Line 284: Difficult to understand. What is the "trained reconstruction"? And what means "the trained reconstruction is able to reconstruct..."?
  - Response: We changed the wording to stress that we mean by a trained reconstruction the neural network with the optimised weights from the training process. The idea is to avoid confusion in the general audience between ocean model and the often used models term in machine learning. We decided to make a clearer statement here and refer in the following manuscript to the trained reconstruction. We changed the wording into (I. 229)"...summarized as learning a set of parameters, represented by a trained reconstruction model (following referred to as trained reconstruction), which uses..."
- Line 303-327: You are saying that if you neglect the spatial info on Argo data, you can utilize a suitable neural network architecture. In the following you say you keep the spatial component using SUSTeR. I do not understand what in the end you do. In addition, understanding SUSTeR and explanations about traffic are not helpful. I suggest to remove this explanation and refer to the publication. Instead please make clearer what you have actually done in the end.
  - Response: We thank the reviewer for this comment. To the first part of the comment, we removed the sentence about the Transformer and LSTM to avoid confusion about our work. Making it clear that we aim to incorporate both the spatial and temporal components of the virtual Argo floats. We tried to give an example that keeping only the temporal structure would be borderline structured and could be processed by sequential models, making the point that the important spatial structure would still be missing. (I.241) "Argo profiles pose a demanding challenge for the design of neural networks, as neural networks are normally designed to handle structured inputs. For spatio-temporal data points, structured input requires a constant number of measurements from constant locations with a static topology (e.g. a grid structure, or a graph), which is both not true for moving Argo floats."
  - Furthermore, we removed the technical part of the traffic prediction comparison and added a brief intuition of why SUSTER is a good match for the Argo problem. We want to mention the different applications to build a bridge for those who will take a look at the SUSTER paper. (I.260)"SUSTER set out with the goal to handle unstructured traffic observations and find a general representation of city traffic, much like the unstructured virtual Argo profiles with the goal to find a general representations of the ocean state." Lastly, we rewrote how SUSTER works and provide a more intuitive description of the learnable clusters, which can dynamically group the virtual Argo profiles.
- Line 329-- Sect. 3.2: Regarding the training procedure of a NN, it would be interesting to see a "loss curve". Often these loss curves are given for the performance of the model on the training set, during the training, as well as on the validation set (unknown).
  - Response: We value the feedback of the reviewer and discussed that thoroughly. We see only limited benefit in including the training/validation loss curves, as they did not

show an interesting pattern. The training uses the early stopping strategy, stopping with a patience of 10 epochs by a maximum of 50 epochs for each experiment. We think that the manuscript would not benefit from analysing the training process in such a technical way. However, we did so during the development to check for problems during the training process. We added to this Response Letter the training curves (train & validation) for the 11 ensemble members of the seasonal time scale from the experiment in section 4.1:

- Line 374, Eqn. 4: I think the denominator is not Var(y), but the total sum of squares  $\sum (y_i-\sqrt{y})^2$ .
  - Response: We thank the reviewer for pointing this out. The reviewer is correct that in Eq. 4 the denominator should be the total sum of squares, \sum\_{i = 1}^{N} (y\_i \overline y)^2, rather than Var(y). We have corrected this in the revised manuscript.
- Page 16, Fig. 4: I suggest to plot the reconstruction curve (blueish) on top of the ground truth (green) to better see it.
  - Response: We thank the reviewer for this good idea. We changed this and all other plots that contain reconstruction time series accordingly.
- Line 595: I guess you again randomized the Argo input data, not leaving it really out? Please mention in the text.
  - Response: We value the feedback of the reviewer, finding this unclear execution. We added the following sentence to describe the experiment in more detail. (I.489) "We retrained the reconstruction in a neural network without argo-related inputs. This removes the influences of the Argo profiles completely from the reconstruction. " By performing a complete retraining, we decoupled the influence from the Argo profiles completely. In this way, it is not necessary to find a strategy (like randomized input) to remove the features, because it was needed for Figure 7 in which we only decoupled the evaluation influence.

- Line 619: If I understand correctly by reading the full paragraph, the reason for no added value of deep Argo is probably just caused by not having enough training data. Thus the influence of deep Argo stays rather unknown. If that is true, please mention already here.
  - Response: We thank the reviewer for their feedback. We changed the latter part of
    the sentence and added a statement about the missing heterogeneous training data
    for the deeper layers. (I. 510) "Across all timescales, we found that the added value is
    limited by the seen training data, which did not cover heterogeneous ocean states in
    the deeper layers, leaving the possible influence of Deep Argo floats undetermined in
    our training setup."

**Corrections:**

```
- 41: Rewrite this part, which sounds strange "... cables that measurement ..."
- 48: Space missing "... balance(Mc ..."
- 155: Figure ??
- 259: "... an high ..." \rightarrow " ... a high ..." and "... an dedicated ..." \rightarrow "... a dedicated
- 309: (?
- 396: "... due to larger ..."
- 399: "The the ..."
- 400: "... limits the compare ..." → sounds strange, please rewrite.
- 427: "data(Jiang" → space
- 478: "... and the also the ..."
- 536: "brach" ? → branch
- 542: "... due to the ..."
- 543: "... transport(Mc" → space
- 630: Change "We test if the test data lays within the training data and its ..." to "We
investigate if the test data lie within the training data and if its ..."
- 706: "... amounts of diverse ...", delete: "... set of ..."
```

- 736: "... mentioned,the ..."  $\rightarrow$  space
  - Response: We thank the reviewer for carefully pointing out the spelling mistakes and rephrasing suggestions and once again for their careful evaluation and constructive feedback. The revisions have helped us to improve the presentation and methodological description of our work. We hope the changes meet the reviewer's expectations and that the revised manuscript has an improved quality.

---

## Author Comment (AC2)

**Final Author comment for RC2**

This paper investigates an interesting approach to monitoring the AMOC that applies machine learning to derive information from Argo float profiles, and other data. I am not an expert in machine learning and my review is from the perspective of an oceanographer.

The authors find that in the model the machine learning technique can make accurate estimates of the AMOC, however, the amount of training data is much greater than is currently available from real observations. Thus, the only prospect for applying the method to estimate the real AMOC would be to train the method on model data. In the discussion, the authors suggest that models are not sufficiently realistic for this to be done now, though this is not analysed. The work is novel and interesting but the presentation is not always easily understandable and some of the analysis seems to confuse different questions. I recommend a major revision

There are two parts to the paper that I think need to be improved.

- 1). Section 4.3. This should be the most important part of the paper as it focuses on the contribution to the AMOC that depends upon the mooring measurements. However, I found this section confusing and the main variable under consideration was not clearly defined
- a) The "RAPID like AMOC", sometimes also referred to in the manuscript as the "geostrophic AMOC" which is the focus of the analysis in section 4.3 is not clearly defined. On line 132 use the term "interior geostrophic transport", I think this is the most accurate description and it would be better throughout. Labelling it as "AMOC" is misleading.
  - Response: We agree with the reviewer's comment that the interior geostrophic transport is not well defined in our manuscript. We replaced all occurrences of geostrophic or RAPID-like AMOC with "interior geostrophic transport". We kept the 'RAPID-like' to highlight that the transport is not a sum of gridboxes in an ocean model. Furthermore, we have added an appendix (Appendix A) which describes in detail how we used the information in McCarthy et al. (2015) to calculate the interior geostrophic transport. We ask the reviewer to share their opinion on whether this should stay in the appendix or be moved into the published code repository. We are convinced that this description will make the manuscript more comprehensible.
  - We validated our approach by applying the interior geostrophic transport calculation from Appendix A to the published RAPID moorings dataset "ts\_gridded.nc" and compared the time series to the published upper mid-ocean transport from the "moc\_transports.nc" dataset. We expect the main difference between the time series to the western boundary wedge. We find that the difference has an average of 1.75Sv and a standard deviation of 3.49Sv. McCarthy et al. (2015) states "The array measures components of the Antilles Current and the Deep Western Boundary Current in combination from Abaco Island to WB2 (WB3) with a mean strength of 1 (4) Sv with a standard deviation of 3 (10) Sv. ". Overall, these statistics validate our calculations and we suspect the small error is due to either non-published or from us not implemented assumptions in the RAPID calculation.

- b). The AMOC is usually defined as the maximum of the overturning stream function so the text on line 251 should be "strength of the stream function at the grid box closest to 1000m". Then on line 255 "we also use an interior geostrophic transport time series".
  - Response: We thank the reviewer for their feedback and have made changes to the
    text to improve its precision. We incorporated the suggestions of the reviewer and
    decided to replace total AMOC with only AMOC throughout the manuscript, based on
    the reviewer's comment in 1a), where we dropped the usage of the geostrophic
    AMOC. To overview all changes we will add the track changes version to our next
    manuscript upload.
- c) Note too that the RAPID "upper mid-ocean time series" usually includes the western boundary wedge. Smeed et al 2018 presented only the geostrophic part east of the mooring WB2 and referred to that as "gyre recirculation".
  - Response: We agree with the reviewer that the former version of our manuscript lacked distinction between the RAPID upper mid-ocean time series and our interior geostrophic transport. We changed the text accordingly. In the beginning of section 4.3 we made clear that it is only part of the umo "...can help to effectively reconstruct the interior geostrophic transport of the RAPID upper mid-ocean component..." (I. 450). When describing the interior geostrophic transport, we explicitly stated that we exclude the WBW and removed the reference to the upper mid-ocean. We changed in line 2123 "For the interior geostrophic transport, we calculate the RAPID upper mid-ocean transport and exclude the western boundary wedge, therefore this is similar to the gyre recirculation in Smeed et al. (2018). To be as similar as possible to the RAPID interior transport, we took the RAPID calculation for the RAPID moorings (McCarthy et al., 2015) and mapped this to the VIKING20X output to share as many assumptions as possible."
- d) On line '536' it is stated that "the RAPID-like geostrophic AMOC, mainly represents the southward deeper return brach of the AMOC" . I think this is incorrect, but the variable is not defined so I am not sure. Normally the southward deep transport should be equal to the AMOC
  - Response: We thank the reviewer for this finding and agree that "southward deeper return branch" is the wrong phrase for what was meant. We changed this sentence in

line 450 into "The contribution of RAPID like interior geostrophic transport we find is to be predominantly negative transport."

- e) When calculating geostrophic transport it is necessary to choose a reference velocity at one level. How was this done in this case? In the RAPID calculation this is done so that the total net transport is zero, so the reference velocity is also influenced by the Ekman and Florida Straits transports.
  - Response: We thank the reviewer for this comment and want to connect this to our answer for point 1a). We add the information about the reference level in the appendix describing the calculation of the interior geostrophic transport. As we are not calculating the total RAPID timeseries we do not enforce a total net zero transport but rather choose reference levels. The velocity calculated from the vertical shear of northward velocity is the surface. Furthermore, we decided to take -4820 meters as level of no transport (see Appendix A Eq. A4) because as mentioned in section "The external transport:" in McCarthy et al. 2015 this is the depth of the deepest RAPID instruments. A deeper level of no transport would include the Antarctic Bottom Water transport which is not part of the RAPID array.
- f) I did not understand why a reduced sampling near the surface to mimic the RAPID observations was done. Surely we want to know how well the ML reconstruction can estimate the actual geostrophic transport? How the missing data from the moorings affects the RAPID estimate is interesting but separate question. The analysis is confounding two different things. For this paper it would be better to focus only on the ML technique.
  - Response: We thank the reviewer for questioning the sampling of missing data for virtual moorings. But we would like to keep this decision because our work aims at the academic question, whether machine learning could be a helpful method to incorporate real-world Argo floats in real-world basin-wide array measurement campaigns aimed at the quantification of the AMOC? The goal is to mimick the challenge an ML approach using real-world data would face as closely as possible. Hence, we use the approx. same data quality (therefore missing mooring data) and only change the amount of data (using ocean models) for the ML training process. Otherwise, we would need to estimate the error that is made by the missing mooring values, which would give this work a flavor of a model vs RAPID comparison. We would like to avoid such a direction and as the reviewer said, focus on the ML technique.
  - We want to show how the missing data in the near-surface of the RAPID moorings limits the availability of measurements. The left plot show the RAPID mooring west from the downloadable data product 'ts\_gridded.nc' from the RAPID website. The white areas show non-available measurements. On the right hand side we show how we mimic this distribution of missing data in the virtual moorings that are extracted from an ocean simulation.

- 2) Section 4.2. "Importance of individual components for the AMOC reconstruction"
- a) This section seems to be confusing two questions. The first question is what components of the circulation contribute most to AMOC variability and the second is which data is most useful for the ML reconstruction.
  - Response: We thank the reviewer for this comment. We fully agree that the first
    question regarding the contribution of the different components to AMOC variability is
    not within the scope of this paper. See our response to 2.b) for details.
- b) There are already quite a few papers that have discussed the first question. In particular Moat et al 2020 discuss how Ekman transport is important at short timescales and that at long time scales most variability is from the mid-ocean transport (see their Figure 2). So the results in Figure 7 do not seem surprising
  - Response: We thank the reviewer for this comment and agree with their perspective. Our goal was to show that we are in line with the current research. The fact itself is, for sure, not surprising. We want to highlight that our data-driven reconstruction utilizes the expected correlations, shown by the previous papers, to give some validation of our method. We rewrote the text to reduce the confusion to this point, explicitly stating that the finding itself is not surprising but necessary for validation. We added the following two sentences after the second paragraph: (I. 426)"On short time scales the high importance of zonal wind stress is not surprising and agrees with current literature (Moat et al. 2020). However, a high importance of wind stress on shorter time scales partially validates the learned AMOC reconstruction, as it utilizes expected and known correlations."
- c) It would be much more interesting if the authors instead examined how much different data contributed to the interior geostrophic transport. Is the surface stress or the Florida Straits transport contributing to the skill in the reconstruction of this component?
  - Response: We thank the reviewer for this valuable feedback. We executed the same experiment as in section 4.2 with the interior geostrophic transport. The result figure is attached to this response. The importance of wind stress is constantly the lowest across all time scales, as expected. Most of the importance is assigned to the virtual Argo profiles for all time scales. A smaller artifact can be found in the seasonal scales when the importance of the Florida Current is larger than the Antilles Current's

importance. Our interpretation is that the Florida Current has a strong seasonal signal, which is evident in a correlation coefficient of -.45 between the seasonal highpass filtered interior geostrophic transport and the Florida Current transport (consistent with Frajka-Williams et al., 2016). Hence, from Florida Current Transport, the algorithm may infer the season which is not explicitly provided via any inputs. In our opinion, however, the manuscript would not benefit from adding these results, also considering the manuscript's length, as a figure or as an appendix. We decided to add the following sentence to the end of section 4.2: (I. 444)"When calculating the feature importance with regard to the interior geostrophic AMOC, not shown in this manuscript, we observed a similar pattern, the Florida Current gains importance for seasonal scales, due to the strong correlation to the AMOC (Frajka-Williams et al. 2016). Despite this, the importance for all other inputs is constant, for the interior geostrophic AMOC across all time scales with virtual Argo profiles being the most important."

**Other comments:**

I found the paper quite long (740 lines excluding figures, tables, references and the abstract) and there are many places where the text could be shortened. E.g. in the introduction "Zilberman et al. (2020) grouped Argo profiles into 6°×6° cells in the Pacific to create a uniform coverage of Argo profiles which could be used for further computation" seems tangential and could be removed. Is it necessary to say (about Argo floats) "Data are transmitted through a satellite connection while the float drifts at the surface for a few hours."? Shortening the text will make the manuscript easier to read.

Reponse: We thank the reviewer for their comment and agree that the manuscript
was at some parts overextended. We removed the mentioned parts in the
introduction (Zilberman, Transmission of Argo signals), also we removed the Deep
Argos from the introduction and shortened the RAPID section. We sharpened the

result section especially in section 4.1 by removing what we considered non-essential information. Overall, we were able to reduce the manuscript to ~620 and believe that this improved its quality. We add a track changes version of the manuscript in next upload to explicitly show the removed parts.

line 115 I do not understand "we also use positions of the RAPID moorings for information about the deeper layers."

Response: We changed the text to highlight our experiment (see section 4.4) about
the influence of measurements deeper than the Argo profiles. The sentence and the
following were meant to describe the three possible options, including no deeper
measurements, including the deeper part of the RAPID moorings, or assume that all
Argos would be Deep Argos ranging up to 6000m depth.

**line 155 "Figure ??"**

• Response: We thank the reviewer for finding this, and we fixed the reference.

line 193 please provide a citation for "graph data structure". Many readers, like me, will not be expert in the techniques of machine learning and so citations are particularly important. Similarly for "explainable AI (X-AI) techniques" on line 485

Response: We thank the reviewer for this helpful comment. We have added appropriate citations for "graph data structure" (doi: 10.1146/annurev-ecolsys-102209-144718) and "explainable AI (X-AI) techniques" (doi: 10.1145/3561048) in the revised manuscript (I. 159).

line 223. The statement "The reconstruction uses the concatenation of the density values from the Argo profiles for the upper 2000 meters and the derivation of the meridional velocity w.r.t. the depth computed with the RAPID mooring locations as information deeper than 2000 meters". is confusing. A concatenation of density and velocity seems odd.

- Response: We thank the author for this feedback and agree that we missed to describe that the input of a neural network does not necessarily share the measurement units. The learning process is based on the statistics of the input variables and therefore can process differently measured variables as shown in Figure 2 C). We made changes to text explaining briefly how measurements with different units can be processed by the same neural network and reference experiment 4.4 that investigates different data sources below 2000 meters depth.
- Line 187: "Please note that the inputs to our neural network must not have the same physical units, since the learning process of the neural networks is based on the statistics of the input variables and not on their semantics. In section 4.4, we investigate the effectiveness of deep information by first testing the removal of the additional mooring data. Next, we assume that all Argo floats would reach up to 6000 meters."

Line 294 I do not understand what is meant by "For the virtual Argo profiles, the goal is to train an embedding (black box in Figure 2 B)) that maps a set of Argo profiles into a hidden space in where similar ocean states are near each other even though their spatial distribution of observations may be different." What is "an embedding"? I think "in where' should be "in which"

- Response: We understand that the term "embedding" is not a familiar oceanography term and is also not well explained in the manuscript. We provide a thorough explanation of the term embedding with a better reference to a manifold learning survey article (Meilă and Zhang,2024). Therefore we restructured section 3.1 such that it follow the schematic from Figure 2. First we describe why Argo profiles with their variable amount and unstructuredness are a challenge to Neural Networks which expect fixed-sized inputs. Next, we show the potential of an embedding and connect the term to mathematical vector spaces that are connected to the ocean manifold. We think this improved the section and gives a better introduction to the embedding term as it is used in machine learning.
- In this answer letter, we want to give a more technical explanation of what we understand as embedding and which task we want it to fulfill. Given a set of Argo profiles with different locations and times the goal of the processing is to identify a structured numerical representation of the underlying ocean state which is independent from the number of profiles. We choose to represent the ocean state by an embedding which maps said set of Argo profiles onto a space of n-dimensional floating-point vectors. Often, the term embedding is used for the n-dimensional vector, but also for the vector space, and for the mapping. Constructing a suitable embedding has two main aspects. First, the same ocean state observed by different spatial and temporal distributions of Argo profiles should result in the same (i.e. negligible Euclidean distance) embeddings. Second, two embeddings should be similar (i.e. small Euclidean distance) if the corresponding ocean states are similar, too. A successful embedding creates a vector representation of the manifold of ocean states that can be used to generalize for unseen but similar ocean states in the evaluation. However, the semantics of these intermediate vectors are not trivial as the exact mapping from the manifold of ocean states to the vector representation is a data-drivenly learned and non-linear function. Without any outside constraints (see black box in Figure 2B) the content of the vectors is not interpretable. The discrepancy between the degree of freedom of the ocean state manifold and the usually smaller dimensionality of the embedding, which limits the information content, can be understood as a low-pass-filter in the training process, enhancing the generalization of the mapping function to focus the mapping function on the largest variabilities in the manifold.
- We used parts of the upper bullet point in the manuscript in line 249 as continuation of the original second paragraph elaborating on the need for such a fixed-sized representation: "A classical binning comes with many assumptions about shape, distance, and connectivity of the bins. This requires a thorough testing of hyperparameters for classical binning approaches, which may be in the end only valid in the study area. However, we aim in this work for a data-driven mapping function which identifies a structured numerical representation (n-dimensional vector) of the underlying ocean state which is independent from the coverage of Argo profiles. Often, the term embedding is used for such a n-dimensional vector, but also for the vector space, and for the mapping (Meilă and Zhang,2024). Constructing a suitable embedding (see black box in Figure 2B) has two main aspects. First, the same ocean state observed by different spatial and temporal distributions of Argo profiles should result in the same (i.e. negligible Euclidean distance) embeddings. Second, two embeddings should be similar (i.e. small Euclidean distance) if the corresponding ocean states are similar, too. A successful embedding creates a

vector representation of the manifold of ocean states that can be used to generalize for unseen but similar ocean states in the evaluation."

Table 1 In the last line I suppose "WS" should be "ZW"?

• Response: Thanks for finding this mistake. It was an relict from an older version, we changed from "WS" to "ZW".

Line 354 is the naming of "test, validation, and training periods" standard? Often "test" and "validation" have similar meaning.

Response: The naming is the standard in the field of machine learning. We
understand that we missed an explanation and added a clarifying sentence. In short,
the validation set is excluded from the training and is used to verify the choice of
hyperparameters, while the test data is only used for the final evaluation.

---

## Author Response (AR2)

**Response Letter (Minor Revisions)**

We thank the reviewers and the editor gracefully for their second round of comments and are glad that our manuscript has improved since the initial submission.

**Comments from the Editor:**

Please also add a proper data citation / reference to your list -- currently you state in the Dava Availability section: "The Argo data were collected and made freely available by the International Argo Program and the national programs that contribute to it. (https://argo.ucsd.edu, https://www.ocean-ops.org https://doi.org/10.17882/42182#110199). " -- in addition, this needs to be a proper citation, as stated on that website, as stated there "Please use the same DOI and citation as the global Argo data snapshot." -- under "how to cite" they provide the correct reference/citation: In the text "Argo (2024)", and in the reference list "Argo (2024). Argo float data and metadata from Global Data Assembly Centre (Argo GDAC) - Snapshot of Argo GDAC of May 09st 2024. SEANOE. https://doi.org/10.17882/42182#110199".

- **Response**: We thank the editor for their comment. We added the reference to Argo snapshot accordingly and cited this reference in the introduction as well as in the data availability statement.

**Comments from Reviewer #1:**

On L186 it is stated that: "Additionally, we provide the stream function below 2000 meters calculated from virtual RAPID moorings as an input to the reconstruction. Presumably, this is the term "MO" in table 1. It is not clear what data was used to calculate "the stream function". I presume only the data below 2000m was used to calculate the streamfunction?

- **Response**: We understand the comment of the reviewer as more details are missing at this point in the manuscript. For these virtual moorings, which indeed are the 'MO' in table 1, we use the same level of no motion (4820 meters) as for the IGT, and aggregate the transport from this level up to 2000 meters. This ensures that no information shallower than 2000 meters is included in the virtual mooring input. We changed the manuscript accordingly by adding the following sentence: "For the calculation of the stream function we used a level of no motion at 4820 meters, similar to the RAPID calculation, and integrated the transport from this bottom level up to 2000 meters depth." (Line 185)

Other minor comments:

- **Response**: We thank the reviewer for their careful reading. We have changed all minor comments accordingly. We will address the following non-trivial changes from the minor

comments:

- In section 4.4 maybe useful to say that even if deep mooring data does not improve the AMOC estimate in these experiments we do expect that it has useful information about the stream function at deeper levels
  - **Response**: We agree with the reviewer that highlighting this hypothesis is reasonable and will be a better wrap-up to this section. The manuscript already contained some parts of this idea in the sentence (Sec 4.4. Line 523): "These findings suggest that while deeper observations can improve reconstructions, their benefit depends on having sufficient representation in the training data.", which we extended now with a second sentence making this more explicit, as suggested by the reviewer: "Although not shown in this experiment, we do expect the deeper observation to contain useful information about the stream function at depth."(L. 524)
- There is an error in equation (A4). I think that "1000" in the last term on the right should be "0".
  - **Response**: We thank the reviewer for their comment, as we agree this equation contains a typo. But what we meant by this equation was to establish a layer of no transport at 4820 meters. Therefore, the right term should be $V(-4820m)|_{0m}$. During the discussion among the authors, we found that we could increase the readability of the equations if we use a positive upward direction, and explicitly distinguish between the reference level for velocities $Z_M$ and the level of no transport $Z_T$. This leads us to the more general set of equations:

$$v(z)|_{Z_M} = \int_z^{Z_M} \frac{\partial v}{\partial z_i}\, \mathrm{d}z'$$

$$V(z)|_{Z_M}^{Z_T} = \int_z^{Z_T} L_x(z')v(z')|_{Z_M}\, \mathrm{d}z'$$

Calculating the IGT at 1000 meters depth, a level of not motion at the surface, and a level of no transport at 4820 meters, with the general equation above, shows why the right term should be $V(-4820m)|_{0m}$ :

$$
\begin{aligned}
V(-1000m)|_{0m}^{-4820m} &= \int_{-1000m}^{-4820m} L_x(z')v(z')|_{0m}\, \mathrm{d}z' \\
&= -\int_{-4820m}^{-1000m} L_x(z')v(z')|_{0m}\, \mathrm{d}z' \\
&= \int_{-1000m}^{0m} L_x(z')v(z')|_{0m}\, \mathrm{d}z' - \int_{-4820m}^{0m} L_x(z')v(z')|_{0m}\, \mathrm{d}z' \\
&= V(-1000m)|_{0m}^{0m} - V(-4820m)|_{0m}^{0m}
\end{aligned}
$$

For the manuscript we do not add the general form because the reference level of the velocities is set to the surface in all our calculations. Nevertheless, we swapped the boundaries of the integral to ensure a positive upward direction and a positive transport in the northward direction for the upper branch. The new equations in the

manuscript are:

$$v(z)|_{0m} = \int_z^{0m} \frac{\partial v}{\partial z_i} \, \mathrm{d}z'$$

$$V(z)|_{0m} = \int_z^{0m} L_x(z')v(z')|_{0m} \, \mathrm{d}z'$$